

# Comparison of performance of tile drainage routines in SWAT 2009 and 2012 in an extensively tile-drained watershed in the Midwest

Tian Guo[1,*], Margaret Gitau[1], Venkatesh Merwade[1,2], Jeffrey Arnold[3], Raghavan Srinivasan[4], Michael Hirschi[5], Bernard Engel[1]

[1]Department of Agricultural & Biological Engineering, Purdue University, West Lafayette, IN 47907, USA
[2]Lyles School of Civil Engineering, Purdue University, West Lafayette, IN 47906, USA
[3]USDA-ARS, Grassland Soil and Water Research Laboratory, Temple, TX 76502, USA
[4]Spatial Sciences Laboratory in the Department of Ecosystem Science and Management, Texas A&M University, College Station, TX 77843, USA
[5]Department of Agricultural & Biological Engineering, University of Illinois Urbana-Champaign, Urbana IL 61801, USA

*Correspondence to*: Tian Guo (guo190@purdue.edu)

**Abstract.** Subsurface tile drainage systems are widely used in agricultural watersheds in the Midwestern U.S. Tile drainage systems enable the Midwest area to become highly productive agricultural lands, but can also create environmental problems, for example nitrate-N contamination associated with drainage waters. The Soil and Water Assessment Tool (SWAT) has been
used to model watersheds with tile drainage. SWAT2012 revisions 615 and 645 provide new tile drainage routines. However, few studies have used these revisions to study tile drainage impacts at both field and watershed scales. Moreover, SWAT2012 revision 645 improved the soil moisture based curve number calculation method, which has not been fully tested. This study used long-term (1991-2003) field site and river station data from the Little Vermilion River (LVR) watershed to evaluate performance of tile drainage routines in SWAT2009 revision 528 (the old routine) and SWAT2012 revisions 615 and 645 (the
new routine). Both routines provided reasonable but unsatisfactory uncalibrated flow and nitrate loss results. Calibrated monthly tile flow, surface flow, nitrate-N in tile and surface flow, sediment and annual corn and soybean yield results from SWAT with the old and new tile drainage routines were compared with observed values. Generally, the new routine provided acceptable simulated tile flow (NSE = 0.50 - 0.68) and nitrate in tile flow (NSE = 0.50 - 0.77) for both field sites with random pattern tile and constant tile spacing, while the old routine simulated tile flow and nitrate in tile flow results for the field site
with constant tile spacing were unacceptable (NSE = -0.77- -0.20 and -0.99 - 0.21 respectively). The new modified curve number calculation method in revision 645 (NSE = 0.56 - 0.82) better simulated surface runoff than revision 615 (NSE = -5.95 ~ 0.5). Calibration provided reasonable parameter sets for the old and new routines in LVR watershed, and the validation results showed that the new routine has the potential to accurately simulate hydrologic processes in mildly-sloped watersheds.

## 1 Introduction

Subsurface drainage systems are common practices in agricultural watersheds in the Midwest area of the US. With subsurface drainage systems, the soil horizontal hydraulic conductivity is increased and makes water drainage from soils to ditches or



subsurface drains effective; the soil vertical hydraulic conductivity is so large that can prevent crop damage from flooding (Mitchell et al.; Guo et al., 2012a; Guo et al., 2012b). In this way, subsurface drainage systems enable large regions of the Midwestern US to become some of the most productive agricultural lands. However, intensive tile drainage systems also create environmental problems, due to contaminants like nitrate-N and pesticides in the water they transport (Guo, 2016). Thus, it is

important to accurately simulate tile drains in hydrological models to correctly predict hydrologic processes and simulate the impacts of land cover and conservation practice changes at the watershed scale.

The Midwestern United States, including Illinois, Iowa, Indiana, Minnesota, Ohio, Michigan, Wisconsin and Missouri, generally have poorly drained soils. These soils remain wet after rainfall events, preventing proper field management. Plant roots are unable to obtain enough aeration in saturated soils, leading to plant growth stress and decreased yields. Consequently,

extensive drainage networks have been built up in the Midwest since 1870 to alleviate the damage caused by uneven drainage (Jaynes and James, 2007). Subsurface drainage allows excess water to leave the soil profile through perforated tubes installed below the soil surface. Water flows into the perforated tubes through cracks between adjacent tiles or holes in the tube and drains away when the water table is higher than the tile. Tile drainage removes surplus water from fields, allows flexible field management and enhances crop production (Sugg, 2007). Tile drainage is widely used in much of the Upper Midwest area.

For instance, over 40,468 km$^2$ (10 million acres) in Illinois have been tiled. Indiana is estimated to have more than 22,000 km$^2$ of land with tile drainage (Sugg, 2007). The Little Vermilion River (LVR) watershed is an extensively tile-drained watershed in Illinois, which is dominated by agricultural lands and with average slope reaching at most 1 % (Zanardo et al., 2012). The LVR watershed has altered hydrology from an extensive subsurface drainage system network, in which the soil vertical hydraulic conductivity is very high and can prevent plant damage from flooding.

Subsurface drainage plays a significant role in water balance in the poorly drained soils of agricultural land, especially in the Midwestern US. For example, Lal et al. (1989) studied tillage-caused alterations in water balance and sediment transport for a corn-soybean rotation in Ohio, and the results demonstrated that the percentage of annual precipitation drained by tiles in plowed conditions and on no-till plots are 33 % to 58 % and 28 % to 59 %, respectively. In terms of water quality, in-stream nitrate loading is particularly influenced by tile drainage. Subsurface tile drainage systems could increase nitrate and pesticide

transport, because they move out of the soil surface and convey soluble nitrate-N from the crop root zone. Nitrate coming from tile drains has been considered the main source of nitrate in rivers and streams in the Midwestern US. Additionally, 89 % - 95 % of nitrate losses in a ditch catchment were transported by the tile drainage system of the catchment (Tiemeyer et al., 2008; Boles et al., 2015). Algoazany et al. (2007) assessed the transport of soluble P through tile drainage and surface runoff and found that crop, discharge and the interactions between sites had significant effects on soluble P concentrations in tile flow,

and annual average soluble P mass loads in subsurface flow was substantially greater than that in surface runoff. Generally, agricultural land with good subsurface drainage would reduce surface runoff, soil erosion and P loss, while increasing nitrate loss. Water discharge and nutrient loads of the Mississippi River has reduced light penetration, and increased aquatic habitat loss and hypoxia in the northern Gulf of Mexico, the largest zone of oxygen-depleted coastal waters in the US (Diaz and Solow, 1999; Rabalais et al., 1999).




The Soil and Water Assessment Tool (SWAT) is a physical based and watershed-scale hydrological model. The SWAT model has been able to simulate tile drainage flow empirically since SWAT2005 (Boles et al., 2015). A new tile drainage simulation method which can accurately describe tile drainage system was used after SWAT2005 (Boles et al., 2015). Specifically, Arnold et al. (1999) improved SWAT2000 with a subsurface flow component and tested the enhanced model (SWAT2002) at a field scale with satisfactory results. However, because pothole impacts had not been included in SWAT2002 and the tile drainage routines were old, the SWAT2002 tile drainage method was not adequate to simulate tile flow and streamflow at a watershed scale  (Arnold et al., 1999; Du et al., 2005). The equation used for tile drainage simulation in SWAT2002 (Neitsch et al., 2011) is:

$$tile_{wtr} = \left(SW_{ly} - FC_{ly}\right) \times (1 - exp[-24/t_{drain}]) \; if \; SW_{ly} > FC_{ly} \; (1)$$

where $tile_{wtr}$ is the amount of water removed from the layer on a given day by tile drainage (mm $H_2O$), $SW_{ly}$ is the water content of the layer on a given day (mm $H_2O$), $FC_{ly}$ is the field capacity water content of the layer (mm $H_2O$), and $t_{drain}$ is the time required to drain the soil to field capacity (hs) (Neitsch et al., 2011).

Du et al. (2005) created an impervious layer and improved the modeling of water table dynamics, and monthly streamflow and tile drainage simulated by SWAT2005 are much better than those simulated by SWAT2002. The time to drain soils to field capacity (TDRAIN) was used to determine the flow rate. Additionally, a new coefficient GDRAIN, the drain tile lag time, was introduced and used as the portion of the flow from tile drains into the streams on a daily basis (Du et al., 2006). Some studies have shown that the tile drainage routine in SWAT2005 could simulate the influence of subsurface drainage on hydrology at a watershed scale (Koch et al., 2013; Sui and Frankenberger, 2008). However, using a drawdown time (TDRAIN) method to simulate tile drains is simplified and limited. Equation (2) (Neitsch et al., 2011) is used for tile drainage simulation in SWAT2005:

$$tile_{wtr} = (h_{wtbl} - h_{drain}/h_{wtbl}) \times (SW - FC) \times (1 - exp[-24/t_{drain}]) \; if \; h_{wtbl} > h_{drain} \; (2)$$

where $tile_{wtr}$ is the amount of water removed from the layer on a given day by tile drainage (mm $H_2O$), $h_{wtbl}$ is the height of the water table above the impervious zone (mm), $h_{drain}$ is the height of the tile drain above the impervious zone (mm), SW is the water content of the profile on a given day (mm $H_2O$), FC is the field capacity water content of the profile (mm $H_2O$), and $t_{drain}$ is the time required to drain the soil to field capacity (hs) (Neitsch et al., 2011).

A new drainage routine which includes the use of the Hooghoudt and Kirkham drainage equations was used to simulate real-world drainage systems more accurately (Moriasi et al., 2005; Moriasi et al., 2012). Based on measured streamflow data from a watershed in Iowa, SWAT with the new tile drain equations was evaluated. The water balance components were simulated, and the results showed that the modified SWAT with the Hooghoudt steady-state and Kirkham tile drain equations simulated flow well (Moriasi et al., 2012). The new tile drainage routines (Eqs. (3), (4) and (5)) added to SWAT2005 are shown below.

When the water table is below the surface and ponded depressional depths are below a threshold, the Hooghoudt steady state equation is used to compute drainage flux:

$$q = 8K_e d_e m + 4K_e m^2/L^2 \; (3)$$



where q is the drainage flux (mm h$^{-1}$), m is the midpoint water table height above the drain (mm), $K_e$ is the effective lateral saturated hydraulic conductivity (mm h$^{-1}$), L is the distance between drains (mm), and $d_e$ is the equivalent depth of the impermeable layer below the tile drains. When the water table completely fills the surface and ponded water remains at the surface for long periods of time, drainage flux is computed using the Kirkham equation (Moriasi et al., 2012; Moriasi et al., 2013):

$$q = 4\pi K_e(t + b - r)/\delta L \quad (4)$$

where t is the average depressional storage depth (mm), b is the depth of the tile drain from the soil surface (mm), r is the radius of the tile drain (mm), and $\delta$ is a dimensionless factor, determined by an equation developed by Kirkham (1957).

When predicted drainage flux is greater than the drainage coefficient, then the drainage flux is set equal to the drainage coefficient:

$$q = DC \quad (5)$$

where q is the drainage flux (mm h$^{-1}$) and DC is drainage coefficient (mm day$^{-1}$) (Moriasi et al., 2012; Moriasi et al., 2013).

These new tile drainage routines have been used to model tile flow at watershed scale since SWAT2009 (Boles et al., 2015). Additionally, the drainage coefficient (DRAIN_CO) was included in the new tile drainage routine in SWAT2012 to control peak drain flow. However, research on simulation of tile flow by the new tile drainage routine is limited. Boles et al. (2015) parameterized the new tile drainage simulation method in SWAT2012 and found that peak tile flow could decrease when moving from SWAT2009 to SWAT2012, because peaks decreased and tiles flowed for a longer period of time. Thus, it is necessary to test and calibrate the new drainage routines in a tile-drained watershed and compare the modelled results by the new tile drainage routines in SWAT2012 with those by the old routines in SWAT2009. Thus, realistic parameters can be selected based on the physical condition, and the impacts of tile drainage on water balance and nutrient loading can be predicted realistically.

SWAT has been widely used to simulate land use change impacts on hydrology and quality (Basheer et al., 2016; Guo et al., 2015; Luo et al., 2012; Shope et al., 2014; Teshager et al., 2016; Wang et al., 2016; Yin et al., 2016), but studies on simulation of tile drainage impact at the watershed scale are few (Arnold et al., 1998). For instance, Sui and Frankenberger (2008) quantified the impact of tile drainage systems on nitrate loss in an extensively tile-drained watershed, and showed that simulated nitrate loss results by SWAT2005 could be used for simulation of nitrate reductions at the watershed scale. Moriasi et al. (2012) used the new tile drain equations in SWAT to evaluate hydrology of a watershed in Iowa and determined value ranges for the new tile drainage parameters, finding that Hooghoudt steady-state and Kirkham tile drain equations could be alternative tile drain simulation methods in SWAT. Boles et al. (2015) tested a new tile drainage routine in a watershed in Indiana using SWAT and found that the new tile drainage routine in SWAT2012 has the potential to predict tile flow and nitrate transported by tiles. Since tile drainage has impacts on hydrology and nutrient loads at the watershed scale, it is important to accurately simulate tile drains in hydrological models to correctly predict hydrologic processes and simulate impacts of land cover and conservation practice changes at the watershed scale. More information about application of realistic parameters for SWAT2012 tile drainage is needed.



Therefore, the main objective of this study is to compare simulated flow, tile flow, runoff, nitrate in tile flow and sediment load results for the new tile drainage routines in SWAT2012 and the old one in SWAT2009 in the LVR watershed and determine which routine provides a better model fit with observed values. We calibrated and validated SWAT models with the new and old tile drainage routines to simulate tile flow and nitrate in tile flow at subsurface stations, surface runoff, and

sediment and nitrate in surface runoff at surface stations, and streamflow, and sediment and nitrate in streamflow at the river station, and compared their performance. We also considered the new tile drainage routine with the improved curve number calculation method. We then determined which tile drainage routine can provide a better model fit. This can allow selection of the most appropriate tile drainage routine suitable for modelling mildly-sloped watersheds in the Midwest with subsurface drainage systems.

## 2 Materials and methods

### 2.1 Study area

The LVR watershed (Fig. 1) is located in east-central Illinois and drains approximately 518 km$^2$. Eighty five percent of the watershed area is in eastern Vermilion County, 13 % of the watershed is in Champaign County, and 2 % of the watershed is in Edgar County. The LVR watershed consists of flat topography, with elevations ranging from 235 meters in the headwaters

to 174 meters at the watershed outlet and with average slope reaching at most 1 % (Zanardo et al., 2012) . The long-term (1991-2000) average annual precipitation for the watershed is 990 mm yr$^{-1}$ (Kalita et al., 2006).

The watershed was subdivided into two subwatersheds, the upstream contributing areas of Georgetown Lake and the LVR. Ninety percent of the LVR watershed is agricultural land used for corn and soybean production, and the remainder consists of grassland, forest land, farmsteads and roadways (Kalita et al., 2006). Annual area planted to soybeans is equal to

the area for corn planting (Algoazany et al., 2007). The dominant soil associations in the LVR watershed are Drummer silty clay loam and Flanagan silt loam (Zanardo et al., 2012; Keefer, 2003), and the dominant hydrologic soil groups are B and C.

The LVR watershed is a typical tile-drained watershed in Illinois. Water quantity and quality data for this watershed are available from a long-term (1991-2003) monitoring project through which data were collected from several subsurface stations, surface stations, river stations and wetland sites in the watershed (Mitchell et al.; Kalita et al., 2006). Based on long-term field

observation data (1991-2000) from the watershed, Mitchell et al. ( and Kalita et al. (2006) studied hydrology of flat upland watersheds in Illinois and demonstrated that the water could remain ponded on the soil surface until it would evaporate, seep or flow to the subsurface when the precipitation rate is higher the infiltration rate of rainfall events, and surface runoff could flow into the streams directly during extremely large rainfall events.

### 2.2 Sites and data for model setup

Two subsurface stations (B and E), two surface runoff stations (Bs and Es), and one river station (R5), with drainage areas of 0.03, 0.076, 0.03, 0.023, and 69 km$^2$, were selected for this study (Fig. 1 and Table 1). Subsurface sites B and E were close to




surface stations Bs and Es, respectively. B and E had similar land use, cropping systems and tile drainage systems with Bs and Es, respectively (Table 1). Elevation, soil, land use and weather data were used for SWAT model setup (Table 1). Daily water discharge data are available at subsurface, surface runoff, and river stations monitored by the Illinois Agricultural Experiment Station, University of Illinois at Urbana-Champaign. Water samples were obtained bi-weekly, while additional samples were

taken during increased flow (Kalita et al., 2006). Daily nitrate and sediment load was computed by multiplying water discharges with nitrate concentration (Yuan et al., 2000). Nitrate and sediment concentration data were not available for every day that water discharge occurred, and available data contained more water discharge than nitrate and sediment concentration measurements (Yuan et al., 2000). Nitrate and sediment loads were computed by multiplying the concentration at a specific time by half the flow volume since the last concentration measurement plus half the flow volume from the concentration

measurement to the next concentration measurement (Kalita et al., 2006; Yuan et al., 2000).

Daily tile flow, surface runoff, nitrate load in tile flow, surface runoff, and streamflow, and sediment load in surface runoff and streamflow were aggregated into monthly data and adopted in this study for model calibration and validation (Table 1). Other stations were not considered due to the quality of their data (Zanardo et al., 2012). Corn and soybean planting, harvest and tillage practice data were collected from landowners (Table 1).

**2.3 Modification to the soil moisture retention parameter calculation method**

The tile drainage routine based on drawdown time in SWAT2009 Revision 528 (Rev.528) was called the "old routine" in this study. The tile drainage routine based on the Hooghoudt and Kirkham equations with a DRAIN_CO in SWAT2012 Revision 615 (Rev.615) was called the "new routine" in this study. SWAT Revision 645 (Rev.645) added a retention parameter adjustment factor (R2ADJ) to Rev.615 to modify the soil moisture retention parameter calculation method (Eqs. (6) and (7))

(Neitsch et al., 2011).

$$S = 25.4(1000/CN - 10) \text{ (6)}$$

$$S = S_{max}(1 - SW/[SW + exp(w_1 - w_2 * SW)]) \text{ (7)}$$

Where S is the retention parameter for a given day (mm), CN is the curve number for the day, $S_{max}$ is the maximum value the retention parameter can achieve on any given day (mm), SW is the soil water content of the entire profile excluding the amount of water held in the profile at wilting point (mm $H_2O$), and $w_1$ and $w_2$ are shape coefficients.

$$w_2 = \left(Log(SW_{fc}/rto3 - SW_{fc}) - Log(SW_{sa}/rtos - SW_{sa})\right)/(SW_{sa} - SW_{fc}) \text{(8)}$$

$$w_1 = Log(SW_{fc}/rto3 - SW_{fc}) + (SW_{fc} \times w_2) \text{ (9)}$$

rto3 is the fraction difference between CN III and CN I retention parameters, rtos is the fraction difference between CN = 99 ($CN_{max}$) and CN I retention parameters, $SW_{fc}$ is amount of water held in soil profile at field capacity, $SW_{sa}$ is amount of water

held in the soil profile at saturation.

In Rev.645, R2ADJ was used to modify shape coefficients, w1 and w2, to increase S and thus decrease CN. R2ADJ ranges from 0 to 1 (Eqs. (10), (11) and (12)). When R2ADJ is 0, CN II is calculated when soil water content is at field capacity.





When R2ADJ is 1, CN II is calculated when soil water content is at saturation. In this case, CN is decreased gradually based on soil from capacity to saturation, which is more reasonable than decreasing CN directly. In reality, CN II could be calculated when soil water content is near saturation (CN II < 100) rather than exactly at saturation (CN II = 100) (Neitsch et al., 2011).

$$MSW_{fc} = SW_{fc} + R2ADJ \times (SW_{sa} - SW_{fc}) \quad (10)$$

$$w_2 = \left(Log(MSW_{fc}/rto3 - MSW_{fc}) - Log(SW_{sa}/rtos - SW_{sa})\right)/(SW_{sa} - MSW_{fc}) \quad (11)$$

$$w_1 = Log(MSW_{fc}/rto3 - MSW_{fc}) + (MSW_{fc} \times w_2) \quad (12)$$

$MSW_{fc}$ is the modified amount of water held in the soil profile at field capacity, and R2ADJ is the newly added retention parameter adjustment factor.

## 2.4 Model setup

SWAT2012 in conjunction with ArcGIS10.1 was used to simulate the LVR watershed. The 30 m Digital Elevation Model (DEM) was used to generate a stream layer for the LVR watershed into the simulation, and subbasins in the LVR watershed were delineated. Landuse data (NLCD 2006) for the study area was obtained from USGS. The National Map Viewer and SSURGO from USDA Web Soil Survey were added into ArcSWAT (Table 1). HRUs were defined using the following thresholds: 0 % landuse, 10 % soil and 0 % slope.

Daily precipitation data from rain gauge stations at sites B, E and 6 km southeast of site R5 were added in ArcSWAT and used for simulation at sites B and Bs, sites E and Es, and site R5, respectively (Table 1). Daily maximum and minimum temperatures, solar radiation, wind speed and relative humidity data from an Illinois State Water Survey (ISWS) station (Champion Station, Latitude: 40.08°, Longitude: -88.24°, Elevation: 219m) closest to the LVR watershed were used (Table 1).

Management operation data for corn and soybean growth at sties B and E were collected (Table 1). Fertilizer was applied 10 days before planting at the rates of 218 kg ha[-1] for anhydrous ammonia and 67 kg ha[-1] for $P_2O_5$. Atrazine was applied at 2.2 kg ha[-1] three days before planting during corn growing years. $P_2O_5$ fertilizer was applied at 56 kg ha[-1] 14 days before planting during soybean production years.

Tile drainage area was determined in HRUs where corn or soybeans were the current land use, slope was lower than 5 %, and soil drainage was somewhat poorly drained, poorly drained, or very poorly drained (Boles et al., 2015; Sugg, 2007; Sui and Frankenberger, 2008), and tile drained area of the LVR watershed is about 75 %.

## 2.5 Parameter adjustments before model calibration

Plant growth parameters for corn and soybean growth simulation at sites B and E were adjusted. Radiation-use efficiency (BIO_E) and harvest index for optimal growing conditions (HVSTI) values for corn growth ranged from 32 to 39, and from 0.41 to 0.54, respectively, based on various studies (Edwards et al., 2005; Kiniry et al., 1998; Lindquist et al., 2005). For



soybean growth, BIO_E and HVSTI values ranged from 13.2 to 25.2, and from 0.44 to 0.59, respectively (Edwards and Purcell, 2005; Mastrodomenico and Purcell, 2012; Sinclair and Muchow, 1999).

The plant growth parameters for corn and soybean growth simulation of sites B and E were adjusted (Table 2). Cibin et al. (2016) adjusted BIO_E and potential heat units (PHU) for corn growth, and PHU, minimum temperature for plant growth
(T_BASE), HVSTI, normal fraction of phosphorus in yield (CPYLD) for soybean growth (Table 2) to reasonably simulate corn and soybean yields for two watersheds in the Midwest US. This study adopted the same adjustment for corn and soybean growth simulation.

Tile drainage simulation parameters were adjusted for the new routine. For Rev.615 and Rev.645, tile depth ranged from 1.05 m to 1.1 m at various sites (Drablos et al., 1988; Singh et al., 2001), and tile depth (DDRAIN) was set as 1.075 m in the
model. The maximum depressional storage selection flag/code (ISAMX) was used to control the method used to calculate the static maximum depressional storage parameter (SSTMAXD), representing the surface storage. When ISMAX is 0, SSTMAXD is allowed to be defined by the user, while when ISMAX is 1, SSTAMXD is dynamically calculated based on rainfall and tillage practices (Moriasi et al., 2005; Moriasi et al., 2012). In this study, ISMAX was set as 0 and SSTMAXD was set as 12 mm, based on previous DRAINMOD (Skaggs et al., 2012) and SWAT studies (Boles et al., 2015). DRAIN_CO,
the amount of water drains in 24 hs, was set as 20 mm day$^{-1}$, describing the size of the main collector drain pipes and the outlet (Sui and Frankenberger, 2008).

**2.6 Model calibration and validation**

Rev.528, Rev.615 and Rev.645 simulated tile flow at sites B and E were compared with the observed values to evaluate tile drainage simulation performance of the old and new routine and the new routine with modified curve number calculation
method. Rev.528 and Rev.615 simulated nitrate in tile flow at sites B and E were compared with the observed values to evaluate nitrate in tile flow simulation performance of the old and the new routines. Rev.615 and Rev.645 simulated surface runoff at site Bs and Es were compared with the observed values to evaluate surface runoff simulation performance of the default soil moisture based curve number calculation method and modified curve number calculation method. Rev.528 and Rev.645 simulated flow at site R5 were compared with the observed values to evaluate flow simulation performance of the
old and new routine. Rev.645 was not used for flow simulation at river station R5, because Rev.645 could not run successfully for the mainly tile drained river station R5. This was thought to be because depth to impervious layer (DEP_IMP) values were too low and the impervious layer was too close to the soil profile, which may have affected the functionality of Rev.645 in simulating ground water and tile flow on a watershed level.

The model was run for a total of 19 years (1985-2003). The first five years (1985-1990) were for model warm-up. Model
outputs, annual corn and soybean yield from 1991 to 1997, and from 1998 to 2003 at sites B and E were compared with the observed values for model calibration and validation, respectively. Monthly tile flow and nitrate in tile flow from 1992 to 1997 and from 1998 to 2003 at site B were compared with the observed values for model calibration and validation, respectively. Monthly tile flow and nitrate in tile flow from 1991 to 1997 and from 1998 to 2002 at site E were compared with the observed



values for model calibration and validation, respectively. Monthly surface runoff, sediment and nitrate in surface runoff at sites Bs and Es, and monthly flow, sediment and nitrate in flow at site R5 from 1993 to 1997 and from 1998 to 2003 were compared with the observed values for model calibration and validation, respectively.

The model was autocalibrated using SWATCUP_5.1.6.2 (SUFI-2). Parameters related to surface runoff, tile drainage,
evapotranspiration (ET), snow, ground water, soil water, sediment losses, and nitrate loss processes were selected during model calibration (Table 2). Ranges of parameters (Table 3) were determined based on previous DRAINMOD studies in the LVR watershed (Singh et al., 2001) and several tile drainage studies in Iowa (Moriasi et al., 2012; Moriasi et al., 2013; Schilling and Helmers, 2008; Singh et al., 2007; Singh et al., 2006; Singh and Helmers, 2008) and Indiana (Boles et al., 2015).

For Rev.528, calibrated values for tile flow simulation parameters at site B, time to drain soil to field capacity (TDRIAN),
drain tile lag time (GDRIAN), and DEP_IMP were used for flow simulation at site R5. For Rev.615, calibrated values for tile flow simulation parameters at site B, DEP_IMP, multiplication factor to determine lateral saturated hydraulic conductivity (LATKSATF), effective radius (RE) and tile spacing (SDRAIN) were modified at site R5, to accurately simulate flow and obtain reasonable water budget results.

**2.7 Model performance evaluation**

Model outputs, annual corn and soybean yield, monthly tile flow and nitrate in tile flow at sites B and E, monthly surface runoff, sediment and nitrate in surface runoff at sites Bs and Es, and monthly flow, sediment and nitrate in flow at site R5 from the old and new routines were compared with observed values for model calibration and validation. Comparison between simulated results from the old and new routines and observed values were plotted. The statistical methods used for verifying model performance included Percent bias/Percent error ($P_{BIAS}$ (%)), the coefficient of determination ($R^2$), the Nash-Sutcliffe
model efficiency coefficient (NSE), the modified NSE (MSE) and the Kling-Gupta efficiency (KGE) (Eqs. (13), (14), (15), (16) and (17)).

$$P_{BIAS} [\%] = (\sum_{i=1}^{n}(Obs - Sim)/\sum_{i=1}^{n} Obs)\times100 \quad (13)$$

$$NSE = 1 - (\sum_{i=1}^{n}(Obs - Sim)^2/\sum_{i=1}^{n}(Obs - \overline{Obs})^2) \quad (14)$$

$$R^2 = \left[\sum_{i=1}^{n}(Obs - \overline{Obs})(Sim - \overline{Sim})\right]^2/\sum_{i=1}^{n}(Obs - \overline{Obs})^2 \sum_{i=1}^{n}(Sim - \overline{Sim})^2 \quad (15)$$

$$MSE = 1 - (\sum_{i=1}^{n}(Obs - Sim)^2/\sum_{i=1}^{n}(Obs - \overline{Obs})^2) \quad (16)$$

$$KGE = 1 - \sqrt{(r-1)^2 + (\alpha-1)^2 + (\beta-1)^2} \quad (17)$$

Where Obs and Sim represent observed and simulated data, respectively. $\alpha = \sigma_{Sim}/\sigma_{Obs}$, and $= \mu_{Sim}/\mu_{Obs}$, and r is the linear regression coefficient between simulated and observed data (Eq. (15)).

Percent bias (Gupta et al., 1999) can measure the average tendency of the simulated data to deviate from the observed data. A
value of 0.0 is optimal for $P_{BIAS}$, representing accurate model simulation. Negative values represent model overestimation bias, and positive values indicate model underestimation bias (Moriasi et al., 2012). If $P_{BIAS} \pm 25$ % for streamflow, $\pm 55$ % for sediment, and $\pm 70$ % for N and P, model simulation results can be considered satisfactory (Moriasi et al., 2007). The $R^2$ value



indicates the strength of the linear relationship between the simulated and observed data. A $R^2$ value of greater than 0.5 is considered reasonable model performance (Moriasi et al., 2007; Guo et al., 2015). The NSE (Nash and Sutcliffe, 1970) can represent how well the plot of observed versus simulated data fits the 1:1 line. The NSE value ranges from $-\infty$ to 1, and the optimal value is 1. A NSE value of greater than 0.5 is considered satisfactory model performance (Moriasi et al., 2007; Guo et al., 2015). A NSE value of 0 means that the simulated values are as accurate as the mean of the observed data, and a negative NSE value represents that the mean value of observed data is a better predictor than the simulated data, meaning unacceptable performance (Moriasi et al., 2007; Melesse and Abtew, 2016). $0.36 \leq NSE \leq 0.72$ and $NSE \geq 0.75$ also have been considered as satisfactory and good simulated data, respectively (Larose et al., 2007; Van Liew et al., 2003; Guo et al., 2015). A modified form of the NSE (Eq. (12)) could decrease the oversensitivity of the NSE to extreme values (Krause et al., 2005), and is sensitive to chronic over- or under predictions. The KGE computes the Euclidian distance of the correlation, the bias, and a measure of variability. The use of KGE (Eq. (13)) improves the bias and the variability measure considerably and decreases the correlation slightly compared to the NSE (Gupta et al., 1999). The KGE value ranges from $-\infty$ to 1. The closer to 1, the more accurate the model is (Melesse and Abtew, 2016). A KGE value of greater than 0.5 is considered satisfactory simulated results (Gupta et al., 1999).

## 3 Results and Discussion

### 3.1 Calibrated parameter values

Parameter ranges and calibrated parameter values for tile flow and nitrate in tile flow simulation at subsurface sites B and E, runoff, sediment and nitrate in runoff simulation at surface sites Bs and Es, and flow, sediment and nitrate in flow simulation at river station R5 were determined (Table 3). Curve number calculation based on soil moisture (ICN = 0) and plant ET (ICN = 1) methods were included in model calibration. For Rev.528 and Rev.615, calibrated curve number (CN2) values ranged from 60 to 65 to accurately simulate surface runoff at field sites, and were reduced by 20 % to accurately simulate streamflow at the river station. These values were reasonable for a watershed dominated by agricultural land based on previous studies (Boles et al., 2015; Moriasi et al., 2012; Neitsch et al., 2011). For Rev.645, calibrated values of newly added curve number calculation retention parameter adjustment factor (R2ADJ) ranged from 0.81 to 0.97 at field sites. In this case CN2 value was calculated when soil water content was near saturation (Eq. (10)), which was reasonable for a mildly-sloped watershed with low runoff (Neitsch et al., 2011). The calibrated parameter sets provide guidance for accurate simulation of tile drainage systems for hydrologic processes at field and watershed scales, and can be used for tile flow, runoff, and sediment and nitrate losses simulation of mildly-sloped watersheds in the Midwest US.

### 3.2 Calibration and validation results for subsurface stations

This section outlines calibration and validation performance for monthly tile flow and nitrate-nitrogen losses for subsurface sites B and E.





### 3.2.1 Calibration and validation results at site B

Simulated annual corn and soybean yields, monthly tile flow, and nitrate in tile flow were compared with observed values during calibration and validation periods at site B (Fig. 2). Model performance in simulating crop yield, tile flow and nitrate in tile flow at site B were evaluated (Table 4).

Performance of the simulated corn and soybean yields from Rev.615 at site B during calibration and validation was satisfactory (Figs. 2a and 2b, and Table 4). Simulated annual corn and soybean yields fit observed values well (Figs. 2a and 2b). $P_{BIAS}$ values of corn and soybean yields during calibration and validation periods were 13 % and 2 %, respectively, indicating fairly accurate model simulation. During the calibration period, $R^2$, NSE, MSE and KGE values for corn and soybean yields were 0.99, 0.91, 0.77 and 0.75, respectively. During the validation period, $R^2$, NSE, MSE and KGE values for corn and

soybean yields were 0.92, 0.91, 0.76 and 0.89, respectively (Table 4). Adjusted crop growth parameters (Table 4) in Rev.615 provided good predictions of corn and soybean yields.

     Performance of the simulated monthly tile flow from Rev.528, Rev.615 and Rev.645 at site B during calibration and validation was satisfactory. Generally, simulated tile flow results for the old routine from Rev.528 were better than those for the new routine from Rev.615 and Rev.645. The modified curve number calculation method in Rev.645 improved surface

runoff simulation and then improved tile flow simulation compared to default curve number calculation method based on soil moisture in Rev.615 (Figs. 2c and 2d, and Table 4). Simulated monthly tile flow was similar to observed values, except that Rev.615 simulated tile flow could not capture tile flow peaks well in May of 1996 and February of 1997 (Fig. 2c). Soil moisture was reduced during long dry periods from June of 1995 to April of 1996. Subsurface tile drains can lower the water table (Sui and Frankenberger, 2008), and long-term water depletion may drop the water table lower than the depth of tiles (1075 mm).

For long-term water table depth simulation (19 years in this study), the computed water table depth may gradually drop as profile soil water decreases due to periods of higher ET, which makes it harder for the water table to rise to the surface after rain events (Moriasi et al., 2013). When water storage is higher than the height of the surface storage threshold (20 % of the static maximum depressional storage (SSTMAXD)) and water table is near the bottom of the soil surface, the Kirkham equation is used to calculate drainage flux (Boles et al., 2015). In this study, overestimation of water table depth might have caused the

new routine not to trigger the Kirkham equation to calculate tile flow drainage even though 1996 was a wet year (annual precipitation was 1008 mm). The new routine in Rev.615 resulted in decreased tile flow peaks and longer storage time (Boles et al., 2015). The new routine in Rev.645 captured tile flow peaks well, although the differences between simulated and observed tile flow values were large in May 1996 and February 1997 (Fig. 2c). This indicates that the newly added curve number calculation retention parameter adjustment factor in Rev.645 calculates curve numbers reasonably well based on the

soil moisture retention curve from field capacity to saturation, and can partition surface runoff and tile flow well. Thus, simulated tile flow results from Rev.645 captured peaks well, and the differences between simulated and observed tile flow values were small after long dry periods (Fig. 2c). $P_{BIAS}$ values of tile flow results were 3 % and 4 % from Rev.528, 14 % and 3 % from Rev.615, and -19 % and -18 % from Rev.645 during calibration and validation periods, respectively, indicating



accurate model simulation. Generally, $R^2$, NSE, MSE and KGE values for tile flow from the three versions were satisfactory ($> 0.5$), except that $R^2$ (0.49) from Rev.615 during calibration period and MSE (0.48) from Rev.645 during validation period were slightly under the acceptable limit (Table 4).

Performance of the simulated monthly nitrate in tile flow from Rev.528 and Rev.615 at site B during calibration and
validation was satisfactory. Generally, simulated nitrate in tile flow results by the old routine from Rev.528 were better than those by the new routine from Rev.615 (Figs. 2e and 2f, and Table 4). Simulated monthly nitrate in tile flow matched observed values well, except that Rev.615 simulated nitrate in tile flow could not capture peaks well in May 1996 and February 1997 (Fig. 2e), which was caused by the failure to predict tile flow correctly during these periods (Fig. 2c). $P_{BIAS}$ values of nitrate in tile flow results were 8 % and 23 % from Rev.528, and 33 % and 18 % from Rev.615 during calibration and validation periods,
respectively, indicating accurate model simulation (Table 4). Generally, $R^2$, NSE, MSE and KGE values for simulated nitrate in tile flow were satisfactory ($> 0.5$). However, $R^2$ (0.37) and NSE (0.22) from Rev.615 during the calibration period were not satisfactory, and MSE (0.43) and KGE (0.48) from Rev.615 during the calibration period were slightly under the acceptable limit (Table 4), which was due to underestimated nitrate values in tile flow after long dry periods (Fig. 2e).

### 3.2.2 Calibration and validation results at site E

Simulated annual corn and soybean yields, monthly tile flow, and nitrate in tile flow were compared with observed values during calibration and validation periods at site E (Fig. 3). Model performance of simulated crop yield, tile flow and nitrate in tile flow at site E were evaluated (Table 4).

Performance of modelled corn and soybean yields from Rev.615 at site E during calibration and validation was satisfactory. Simulated annual corn and soybean yields were similar to observed values (Figs. 3a and 3b, and Table 4). $P_{BIAS}$
values of corn and soybean yields during calibration and validation periods were -2 % and 5 %, respectively, indicating fairly accurate model simulation. During the calibration period, $R^2$, NSE, MSE and KGE values for corn and soybean yields were 0.95, 0.95, 0.80 and 0.95, respectively. During the validation period, $R^2$, NSE, MSE and KGE values for corn and soybean yields were 0.92, 0.88, 0.71 and 0.91, respectively (Table 4). Adjusted crop growth parameters (Table 4) in Rev.615 provided good predictions of corn and soybean yields.

Performance of the modelled monthly tile flow from Rev.615 and Rev.645 at site E during calibration and validation was satisfactory. Generally, simulated tile flow results for the new routine from Rev.615 and Rev.645 were better than those from the old routine from Rev.528. Simulated monthly tile flow from Rev.615 and Rev.645 fit observed values well (Figs. 3c and 3d, and Table 4). However, Rev.528 simulated tile flow was overestimated at tile flow peaks in November 1992, May 1996, March 1997 (Fig. 3c), May and June of 1998, December 2001, and February, April and May of 2002 (Fig. 3d). Rev.528
simulated tile flows were underestimated from May to October in 1992, from June to November in 1994, from July in 1995 to March in 1996 (Fig. 3c), from May in 1999 to February in 2000, from May to August in 2001, and from July to December in 2002 (Fig. 3d). The old routine in Rev.528 has the potential to overestimate tile flow peaks, since simulated tile flow by the old routine was controlled by a simple drawdown time parameter (TDRIAN), and tiles were allowed to carry an unlimited





maximum of water no matter how intense the rainfall. Thus, the old routine overestimated tile flow peaks for site E (Figs. 3c and 3d).

The new routine in Rev.615 and Rev.645 incorporates the DRAIN_CO, and tile flow peaks can be limited by the radius of the tile. In this case, the tiles could flow for a slightly longer period of time, and simulated tile flow matched well with observed values (Figs. 3c and 3d). The old routine was used to simulate tile flow on days when the simulated height of the water table exceeded the height of the tile drain (Neitsch et al., 2011). Tile drainage systems can cause water table recession in tile-drained soil. Water table was lower when respiratory activity was highest in summer (Muhr et al., 2011), which may be lower than the depth of subsurface tiles during long dry summer periods. Water table depth calculation based on change in the soil water for the whole soil profile tended to overestimate the distance between water table and the soil surface when long-term simulations were performed, most commonly in cases where days without rainfall dominated (Moriasi et al., 2013). Thus, Rev.528 simulated tile flow was zero during long dry summer periods. The more physically-based equations and the DRAIN_CO in the new routine in Rev.615 and Rev.645 can reduce the flashiness of the tile flow simulation and result in lower tile flow peak and longer recession (Boles et al., 2015). The new routine in Rev.615 and Rev.645 provided more reasonable tile flow simulation for site E (Figs. 3c and 3d). $P_{BIAS}$ values of tile flow results were -6 % and 12 % from Rev.615 and -17 % and -2 % from Rev.645 during calibration and validation periods, respectively, indicating fairly accurate model simulation. $R^2$, NSE, and KGE values for tile flow from Rev.615 and Rev.645 were satisfactory (> 0.5). However, MSE from Rev.615 (0.28) and from Rev.645 (0.27) during calibration period, and MSE from Rev.615 (0.31) and from Rev.645 (0.34) during validation period were under the generally acceptable limit (Table 4). $P_{BIAS}$ value of tile flow results from Rev.528 during calibration period was -37 %, indicating overestimated model values. NSE, MSE and KGE values from Rev.528 during calibration and validation periods were unacceptable (< 0.5) (Table 4).

Performance of the simulated monthly nitrate in tile flow from Rev.615 at site E during calibration and validation was satisfactory. Generally, simulated nitrate in tile flow results for the new routine from Rev.615 were better than those for the old routine from Rev.528 (Figs. 3e and 3f, and Table 4). Simulated monthly nitrate in tile flow matched observed values well, except that Rev.615 simulated nitrate in tile flow was underestimated in May 2002 (Fig. 3f), which is caused by the underestimation of tile flow during this period (Fig. 3d). Performance of the modelled monthly nitrate in tile flow from Rev.528 for site E during calibration and validation was unsatisfactory (Figs. 3e and 3f), which is likely caused by the failure to predict accurate tile flow (Figs. 3c and 3d). $P_{BIAS}$ values of nitrate in tile flow results from Rev.615 during calibration and validation periods were 26 % and 20 %, respectively, indicating accurate model simulation. $R^2$, NSE, and KGE values for simulated nitrate in tile flow from Rev.615 during calibration and validation periods were satisfactory (> 0.5), but MSE values during calibration (0.30) and validation (0.34) periods were under the acceptable limit (< 0.5) (Table 4). $P_{BIAS}$ values of nitrate in tile flow results from Rev.528 during the validation period was 36 %, indicating underestimated model simulation. NSE, MSE, and KGE values for simulated nitrate in tile flow from Rev.528 during calibration and validation periods were unsatisfactory (< 0.5) (Table 4).





Simulated monthly tile flow results for Rev.615 at sites B and E were better than previous DRAINMOD and Root Zone Water Quality Model (RZWQM) simulated results (Singh et al., 2001), since both DRAINMOD and RZWQM models overestimated daily tile flow at these sites to obtain an acceptable $R^2$ value (> 0.5), but they did not match well with the observed values generally from 1993 to 1998. Simulated monthly tile flow results for Rev.615 at sites B and E were similar to

the observed values, and obtained acceptable $P_{BIAS}$, $R^2$, NSE, MNS and KGE generally from 1991 to 2003.

### 3.3 Calibration and validation results for surface stations

This section describes calibration and validation performance for monthly surface runoff, sediment and nitrate-nitrogen losses at surface sites Bs and Es. The LVR watershed is dominated by agricultural land with extensive tile drainage system. Direct surface runoff was a small percentage (≤ 15 %) of the stream flow in the LVR watershed, and was nearly zero for years 1995

and 1997, even though there was sufficient precipitation (Mitchell et al.). Thus, it is challenging to simulate surface runoff, sediment load, and nutrient load in runoff in the LVR watershed.

### 3.3.1 Calibration and validation results as site Bs

Performance of the modelled monthly surface runoff from Rev.645 at site Bs during calibration and validation was satisfactory. Modelled monthly surface runoff from Rev.615 at site Bs during calibration and validation was unsatisfactory. Generally,

simulated surface runoff results from Rev.645 with the improved curve number calculation method were better than those from Rev.615 with the default soil moisture based curve number calculation method. Simulated surface runoff results from Rev.645 were better than those from Rev.615 for site Bs (Figs. 4a and 4b, and Table 4). Generally, simulated monthly surface runoff from Rev.645 was similar to observed values. Rev.615 simulated surface runoff results were higher than observed values (Figs. 4a and 4b). For Rev.615, calibration ranges of CN2 (-20 % ~ -10 %) and calibrated CN2 value (60.1) were realistic for a

watershed dominated by agricultural land (Table 3), and simulated surface runoff was overestimated (Figs. 4a and 4b). $P_{BIAS}$ values of surface runoff results from Rev.615 during calibration and validation periods were -614 % and -475 %, respectively, representing overestimated simulation results. $P_{BIAS}$ values of surface runoff results from Rev.645 during calibration and validation periods were -26 % and -74 %, indicating slightly overestimated and overestimated simulation results, respectively. Generally, $R^2$, NSE, MSE and KGE values for simulated surface runoff results from Rev.615 were unacceptable (< 0.5) (Table

4). $R^2$, NSE, MSE and KGE values for simulated surface runoff results from Rev.645 were acceptable (> 0.5) (Table 4), except that MSE during calibration (0.48) and validation (0.41) periods were slightly under the acceptable limit, and the KGE value during the validation period (0.18) was unacceptable (Table 4). In this watershed with flat topography and dominated by tile drainage, surface runoff was small for surface station Bs and nearly zero from 1994 May to 1996 March (Fig. 4a) and from 1999 March to 2002 April (Fig. 4b).

Performance of the modelled monthly sediment load in flow from Rev.645 for site Bs was satisfactory during calibration and reasonable during validation (Figs. 4c and 4d, and Table 4). Simulated monthly sediment load from Rev.645 was similar to observed values (Figs. 4c and 4d), except that simulated sediment load was lower than the observed value for March 1999



(Fig. 4d). $P_{BIAS}$ values of sediment load results were -5 % and 37 % from Rev.645, during calibration and validation periods, respectively, indicating accurate simulation results (Table 4). $R^2$, NSE, MSE and KGE values for simulated sediment during the calibration period were satisfactory (> 0.5) (Table 4). $R^2$, NSE, MSE and KGE values for simulated sediment during validation period were unsatisfactory (< 0.5) (Table 4), which was because the simulated sediment could not capture the sediment peak well for March 1999 (Fig. 4d), and performance evaluation methods are sensitive to high values. The magnitude of sediment load for site Bs was small, thus simulated results were reasonable even though simulated sediment load was underestimated for March 1999 (Fig. 4d).

Performance of the modelled monthly nitrate load in surface runoff from Rev.645 for site Bs during calibration and validation was reasonable (Figs. 4e and 4f, and Table 4). Simulated monthly nitrate load was similar to observed values (Figs. 4e and 4f), except that simulated nitrate load values were lower than the observed values in May of 1996 and 1998, and January 1999 (Figs. 4e and 4f). $P_{BIAS}$ values of nitrate load results were 79 % and 53 % during calibration and validation periods, indicating underestimated model simulation. Generally, $R^2$, NSE, MSE and KGE values for simulated nitrate load were unsatisfactory (< 0.5) (Table 4). However, Rev.645 simulated nitrate in surface flow was reasonable, as nitrate in surface runoff was low given the watershed was dominated by tile flow.

### 3.3.2 Calibration and validation results at site Es

Performance of the modelled monthly surface runoff results from Rev.615 and Rev.645 at site Es was satisfactory during the calibration period and unsatisfactory during the validation period. Simulated surface runoff results from Rev.645 were better than those from Rev.615 for site Es (Figs. 5a and 5b, and Table 4). Generally, simulated monthly surface runoff from Rev.615 and Rev.645 fit observed values well during the calibration period (Fig. 5a), and provided higher than observed values during the validation period (Fig. 5b). For Rev. 615, calibrated CN2 value (60.1) was realistic for watersheds dominated by agricultural land (Table 3), and simulated surface runoff was overestimated. $P_{BIAS}$ values of surface runoff results from Rev.615 during calibration and validation periods were -107 % and -143 %, respectively (Table 4), representing overestimated simulation results. $P_{BIAS}$ values of surface runoff results from Rev.645 during calibration and validation periods were -18 % and -99 % (Table 4), indicating slightly overestimated and overestimated simulation results, respectively. Generally, $R^2$, NSE, MSE and KGE values for simulated surface runoff results from Rev.615 were unacceptable (< 0.5), except that $R^2$ values were 0.71 and 0.55 during calibration and validation periods, respectively, and NSE value was 0.50 during calibration period (Table 4). $R^2$, NSE, MSE and KGE values for simulated surface runoff results from Rev.645 were acceptable during the calibration period (> 0.5) and unacceptable during the validation period (Table 4). In this mildly-sloped watershed with extensive tile drainage systems, surface runoff was small for surface station Es and nearly zero from 1994 June to 1995 April (Fig. 5a) and from 1998 July to 2002 March (Fig. 5b).

Performance of the modelled monthly sediment load in flow from Rev.645 for site Es was satisfactory during calibration (Fig. 5c) and reasonable during validation (Fig. 5d). Simulated monthly sediment load from Rev.645 was similar to observed values during the calibration period, except that simulated sediment load was lower than the observed value for May 1996



(Fig. 5c). Simulated monthly sediment load from Rev.645 did not match observed values well during the validation period, and had difficulty in capturing sediment load peaks well (Fig. 5d). $P_{BIAS}$ values of sediment load results were 32 % and 22 % from Rev.645 during calibration and validation periods, respectively, indicating accurate simulation results (Table 4). $R^2$, NSE, MSE and KGE values for simulated sediment during calibration and validation periods were unsatisfactory ($< 0.5$) (Table 4),

except that $R^2$ (0.79) was acceptable and NSE (0.46) was slightly under the acceptable limit during the calibration period (Table 4). Simulated results from Rev.645 were reasonable, even though evaluation statistics were unsatisfactory, as the magnitude of sediment load was small for the mildly-sloped site.

Performance of the modelled monthly nitrate load in surface runoff from Rev.645 for site Es during calibration and validation was reasonable (Figs. 5e and 5f). Simulated monthly nitrate load was similar to observed values during the

calibration period, except that simulated nitrate load values were lower than the observed values in May 1996 (Fig. 5e). Simulated nitrate load in surface runoff could not capture nitrate load peaks well during the validation period (Fig. 5f). $P_{BIAS}$ values of nitrate load results were 25 % and 83 % during calibration and validation periods, indicating accurate and underestimated model simulation, respectively. Generally, $R^2$, NSE, MSE and KGE values for simulated nitrate load were unsatisfactory ($< 0.5$) (Table 4). Rev.645 simulated nitrate in surface flow for site Es was reasonable, as nitrate in surface

runoff was small as surface runoff rarely occurred.

### 3.4 Calibration and validation results for river station R5

Simulated monthly flow, sediment and nitrate load from Rev.528 and Rev.615 were compared with observed values during calibration and validation periods for site R5 (Fig. 6). Model performance of simulating flow, sediment, and nitrate load for site R5 were evaluated (Table 4).

Performance of the modelled monthly flow from Rev.528 and Rev.615 at site R5 during calibration and validation was satisfactory. Simulated monthly flow results from Rev.528 were slightly better than those from Rev.615 at site R5 (Figs. 6a and 6b, and Table 4). Generally, simulated monthly flow was similar to observed values (Figs. 6a and 6b). However, Rev.528 simulated flow values were higher than observed values in May 1996 and December 1997 (Fig. 6a), which was mainly caused by overestimation of tile flow during these periods. Simulated tile flow by the old routine in Rev.528 was controlled by a

simple drawdown time parameter (TDRIAN), no matter how intense the rainfall. Thus, Rev.528 has the potential to overestimate tile flow peaks. Rev.528 and Rev.615 simulated flow values were slightly higher than observed values from June to November of 1994, 1996 and 1998 (Figs. 6a and 6b), which was mainly because of the overestimation of surface runoff during these periods. Calibration ranges of CN2 (-20 % - -10 %) and plant ET curve number coefficient CNCOEF (0.5 ~ 2) were realistic for a watershed dominated by agricultural land (Table 3), and simulated surface runoff was overestimated.

Rev.528 and Rev.615 simulated flow values were lower than observed values from January 2000 to February 2001 (Fig. 6b), which was mainly caused by underestimation of tile flow. Since the water table was lower than the tiles after the long dry period in 1999, the old routine in Rev.528 could not simulate tile flow, and the new routine in Rev.615 could not use the Kirkham equation to calculate tile drainage flux. $P_{BIAS}$ values of flow results from Rev.528 and Rev.615 during the calibration



period were -36 % and -48 %, respectively, representing overestimated simulation results. $P_{BIAS}$ values of flow results from Rev.528 and Rev.615 during the validation period were -1 % and -11 %, respectively, indicating fairly accurate simulation results. Generally, $R^2$, NSE and KGE values for simulated flow results from Rev.528 and Rev.615 were satisfactory (> 0.5), except that NSE (0.48) from Rev.615 during the validation period was slightly under the acceptable limit (Table 4). MSE from

Rev.615 during the calibration period (0.43) was slightly under the acceptable limit, and MSE from Rev.528 (0.36) and Rev.615 (0.26) during the validation period was unacceptable (Table 4). Simulated flow results from Rev.528 and Rev.615 during the calibration period had a better match with observed values (Fig. 6a) and better $P_{BIAS}$, $R^2$, NSE, MSE, and KGE values than those during the validation period. The long dry period during 1999 affected water table depth calculation and then simulation of tile flow from Rev.528 and Rev.615 during 2000 and 2001.

Simulated average annual tile flow values from Rev.528 (128 mm) and Rev.615 (129 mm) were 14 % and 15 % of total precipitation respectively over the period from 1992 to 2003. Simulated average annual ET values from Rev.528 (585 mm) and Rev.615 (571 mm) were 71 % and 69 % of total precipitation, respectively. Simulated average annual water yield values from Rev.528 (248 mm) and Rev.615 (265 mm) were 27 % and 29 % of total precipitation, respectively. Flow partitioning appeared reasonable for simulated results from Rev.528 and Rev.615 based on previous watershed-scale tile drainage

simulation studies (Boles et al., 2015; Moriasi et al., 2012; Moriasi et al., 2013). Major flow paths are important in determining sediment and nitrate loads.

Performance of the modelled monthly sediment load in flow from Rev.528 and Rev.615 at site R5 during calibration and validation was reasonable. Simulated monthly sediment load in flow results from Rev.615 were better than those from Rev.528 at site R5 (Figs. 6c and 6d, and Table 4). Simulated monthly sediment load from Rev.528 and Rev.615 matched observed

values fairly well, except that both routines could not capture sediment load peaks well (Figs. 6c and 6d). This was caused by the failure to predict surface runoff well. $P_{BIAS}$ values of sediment load results were -141 % from Rev.528, and -474 % from Rev.615 during the validation period, respectively, indicating overestimated model simulation during validation (Table 4). Generally, $R^2$, NSE, MSE and KGE values for simulated sediment were unsatisfactory (< 0.5), except for KGE (0.56) from Rev.615 during the calibration period, and $R^2$ (0.76) from Rev.615 during validation which were acceptable (Table 4).

However, the LVR watershed is a mildly-sloped watershed with extensive tile drainage systems, which was dominated by tile flow, and surface runoff and sediment in surface runoff were low, and it was challenging to simulate sediment load accurately. Rev.528 and Rev.615 simulated sediment load had difficulty in matching sediment load peaks (Figs. 6c and 6d), and performance evaluation results were unacceptable generally (Table 4), but simulated sediment load can still be considered reasonable, since the magnitude of sediment load in this mildly-sloped watershed was small (Figs. 6c and 6d).

Performance of the modelled monthly nitrate load in flow from Rev.528 and Rev.615 at site R5 during calibration and validation was satisfactory. Simulated monthly nitrate loads in flow results from Rev.615 were better than those from Rev.528 at site R5 (Figs. 6e and 6f, and Table 4). Simulated monthly nitrate load was similar to observed values, except that Rev.528 simulated nitrate load values were higher than observed values in May 1996, December 1997, and May 2002 (Figs. 6e and 6f), which was mainly caused by overestimation of tile flow during these periods. Rev.528 and Rev.615 simulated nitrate load



values were lower than observed values during June 1997, and May and June of 2002 (Figs. 6e and 6f), which was mainly caused by underestimation of tile flow during these periods. $P_{BIAS}$ values of nitrate load results were 11 % and 31 % from Rev.528 during calibration and validation periods, and 17 % and 37 % from Rev.615 during calibration and validation periods, indicating fairly accurate model simulation. Generally, $R^2$, NSE, MSE and KGE values for simulated nitrate load were

satisfactory (> 0.5). However, NSE (0.33) and MSE (0.40) from Rev.528 during the calibration period were unsatisfactory, and KGE (0.48) from Rev.615 during the validation period were slightly under the acceptable limit (Table 4). $R^2$, NSE and MSE values from Rev.615 were 0.63, 0.48 and 0.26 for simulated flow, and 0.67, 0.58 and 0.50 for simulated nitrate load during the validation period (Table 4), which may be because simulated nitrate load results could capture peaks better than simulated flow results during May 2000 and February 2001 (Fig. 6b).

The new tile drainage routine in Rev.615 was improved compared to the old routine in Rev.528 (Fig. 6, and Table 4). Capacity of water that can be drained by tiles was unlimited by the old routine in Rev.528, and resulted in overestimated flow results during the calibration period (Fig. 6a), while simulation of tile flow from the new routine in Rev.615 incorporated DRAIN_CO to control peak drain flow (Fig. 6a). Rev.528 could not simulate tile flow once the water table was lower than tile depth, while Rev.615 could simulate tile flow by the Hooghoudt equation once the water table dropped after a long dry period

during validation (Fig. 6b). Rev.615 incorporated tile parameters, such as DRAIN_CO, DDRAIN, LATKSATF, RE and SDRAIN to represent characteristics of tile drainage system, which can simulate tile flow more realistically. Some processes in Rev.615 could be improved. For instance, DEP_IMP can represent depth to impervious layer and soil permeability and can be separated in the model. Water table depth calculation can determine which equation will be used for tile flow simulation, and water table depth calculation during long dry periods can be improved to better simulate tile flow.

Limitations of this work include limited observed rainfall data for site R5, water table depth calculation after long dry periods, and difficulty in simulating surface runoff, sediment, and nitrate in surface runoff from this extensively tile drained, mildly-sloped watershed. Observed rainfall data for site R5 was from the closest rain gauge station located 6 km southeast of site R5, which may impact the accuracy of flow simulation. There is an opportunity to improve the representation of tile drainage systems in SWAT, and improve Rev.645 functionality at watershed scales. The new routine and the improved curve

number calculation method can be tested for more individual tiles and watersheds.

## 4 Conclusions

In this study the old tile drainage routine in SWAT2009 (Rev.528) and the new tile drainage routine in SWAT2012 (Rev.615 and Rev.645) were used to simulate monthly tile flow, nitrate in tile flow, surface runoff, and sediment and nitrate in surface runoff at field sites, and monthly flow, sediment and nitrate in flow at a river station. Performance of both routines was

evaluated and compared with observed values.

    The results showed that Rev.615 satisfactorily simulated corn and soybean yields at field sites, and both routines provided satisfactory tile flow and nitrate in tile flow results at subsurface sites, satisfactory flow and nitrate load in flow, and reasonable




sediment load in flow results at the river station after model calibration. Rev.645 with an improved curve number calculation method provided satisfactory surface runoff, and reasonable sediment and nitrate load in surface runoff results at surface stations.

Generally, simulated tile flow results for the old routine were better than those for the new routine at site B, while simulated tile flow results from the new routine were better than those from the old routine at site E. Nitrate in tile flow results from the new routine were better than those from the old routine at both sites. Simulated flow and nitrate in flow results from the new routine were better than those from the old routine at site R5. The new routine provided more realistic and accurate simulation of tile drainage, and the new curve number retention parameter adjustment factor in Rev.645 improved surface runoff simulation, and is suitable for surface runoff simulation in mildly-sloped watersheds.

The results determined which tile drainage routine can provide a better model fit, and provided representative parameter sets in SWAT for simulation of tile flow, nitrate in tile flow, surface runoff, sediment and nitrate in surface runoff at field scale, and simulation of streamflow, and sediment and nitrate in streamflow at watershed scale in tile-drained watersheds. The results provide guidance for selection of tile drainage routines and related parameter sets for tile drainage simulation at both field and watershed scales. It is necessary and important to test tile drainage routines and related parameter sets before their applications in hydrological and water quality modeling.

**Acknowledgements**

The data used in this publication from the Little Vermilion River Watershed were a contribution of the Illinois Agricultural Experiment Station, University of Illinois at Urbana-Champaign as a part of Projects 10-309 and 10-301 and Southern Regional Research Project S-1004 (formerly S-249 and S-273). Supported in part with funds from USDA-CSREES under special projects 91-EHUA-1-0040 and 95-EHUA-1-0123, NRI project 9501781, and Special Project 95-34214-2266 (Purdue sub-contract 590-1145-2417-01). In addition, this work was supported with funds from the Illinois Council on Food and Agricultural Research and with the assistance of the Champaign County Soil and Water Conservation District which sponsored the installation of the County Line gaging station. Faculty from the Department of Agricultural Engineering supervising the collection and reduction of these data were: J. K. Mitchell, M. C. Hirschi, P. Kalita, and R. A. C. Cooke.

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



**Figure captions:**

**Fig. 1. Monitored subsurface, surface and river stations in the LVR watershed.**

**Fig. 2. Calibration and validation results for annual crop yields (a and b), and monthly tile flow (c and d) and nitrate-nitrogen losses in tile flow (e and f) at site B. Obs, Prcp, Rev528, Rev615 and Rev645 represent Observed, Precipitation, Revision 528, Revision 615 and Revision 645, respectively.**

**Fig. 3. Calibration and validation results for annual crop yields (a and b), and monthly tile flow (c and d) and nitrate-nitrogen losses in tile flow (e and f) at site E. Obs, Prcp, Rev528, Rev615 and Rev645 represent Observed, Precipitation, Revision 528, Revision 615 and Revision 645, respectively.**

**Fig. 4. Calibration and validation results for monthly surface runoff (a and b), sediment (c and d) and nitrate-nitrogen losses in surface runoff (e and f) at site Bs. Obs, Rev615 and Rev645 represent Observed, Revision 615 and Revision 645, respectively.**

**Fig. 5. Calibration and validation results for monthly surface runoff (a and b), sediment (c and d) and nitrate-nitrogen losses in surface runoff (e and f) at site Es. Obs, Rev615 and Rev645 represent Observed, Revision 615 and Revision 645, respectively.**

**Fig. 6. Calibration and validation results for monthly flow (a and b), sediment (c and d) and nitrate-nitrogen losses in flow (e and f) at site R5. Obs, Prcp, Rev528, and Rev615 represent Observed, Precipitation, Revision 528 and Revision 615, respectively.**



**Table 1** Monitored subsurface, surface and river stations and data for simulation in the LVR watershed, and cropping and tillage practices for sites B and E

| Monitored subsurface, surface and river stations | | | | |
|---|---|---|---|---|
| Site | Soils | Station | Drainage system | Cropping |
| B | Drummer silt clay loam | Subsurface | Random tile drainage tubing systems in depressional areas | Reduced-Tillage Beans-Corn |
| Bs | Flanagan silt loam | Surface | | |
| E | Sabina silt loam | Subsurface | Complete tile drainage system at 28 m spacing | No-Tillage Corn-Beans |
| Es | Xenia silt loam | Surface | | |
| R5 | - | River | Random tile systems | - |

| Data for tile drainage simulation | | | |
|---|---|---|---|
| Data type | Source | Format | Date |
| Elevation | [1]USGS The National Map Viewer | 30m raster | |
| [2]SSURGO | [4]USDA Web Soil Survey | Polygon Shapefile | |
| [3]LULC | [1]USGS The National Map Viewer | Raster | 2006 |
| Temperature, solar radiation, relative humidity and wind speed | [5]ISWS | Tabular data | 1991 - 2003 |
| Precipitation | [6]UIUC | Tabular data | 1991 - 2003 |
| Corn and soybean yield, planting, harvest, fertilization and tillage for sites B and E | [6]UIUC | Tabular data | 1991- 2003 |
| Tile flow, nitrate-nitrogen in tile flow, site B | [6]UIUC | | 1992 – 2003* |
| Tile flow, nitrate-nitrogen in tile flow, site E | [6]UIUC | | 1991 - 2002 |
| Surface runoff, sediment and nitrate-nitrogen in runoff for sites Bs and Es | [6]UIUC | | 1993 - 2003 |
| Flow, sediment and nitrate-nitrogen in flow for site R5 | [6]UIUC | | 1993 - 2003 |

| Cropping and tillage practices for sites B and E | | | | | | | | |
|---|---|---|---|---|---|---|---|---|
| Year | Crop | | Planting date (Month/day) | | Harvest date (Month/day) | | Tillage type | |
| Site | B | E | B | E | B | E | B | E |
| 1991 | Soybean | Corn | 05/08 | 09/21 | 09/21 | 10/08 | Reduced | No |
| 1992 | Corn | Soybean | 04/30 | 10/06 | 10/06 | 10/06 | tillage- | tillage |
| 1993 | Soybean | Corn | 05/17 | 09/30 | 09/30 | 11/08 | chisel | |
| 1994 | Corn | Soybean | 04/21 | 09/13 | 09/13 | 10/06 | plowed, | |
| 1995 | Soybean | Corn | 06/04 | 10/02 | 10/02 | 10/17 | disked, or | |
| 1996 | Corn | Soybean | 04/18 | 09/19 | 09/19 | 10/17 | field | |
| 1997 | Soybean | Corn | 04/29 | 09/26 | 09/26 | 10/15 | cultivated | |
| 1998 | Corn | Soybean | 04/26 | 09/23 | 09/23 | 09/28 | | |
| 1999 | Soybean | Corn | 05/07 | 09/19 | 09/19 | 11/09 | | |
| 2000 | Corn | Soybean | 04/13 | 09/19 | 09/19 | 10/04 | | |
| 2001 | Soybean | Corn | 04/30 | 09/27 | 09/27 | 10/29 | | |
| 2002 | Corn | Soybean | 05/21 | 10/01 | 10/01 | 10/01 | | |
| 2003 | Soybean | Corn | 05/22 | 10/01 | 10/01 | 10/27 | | |

[1]USGS: U.S. Geological Survey
[2]SSURGO: Soil Survey Geographic Database
[3]LULC: Land Use/Land Cover
[4]USDA: U.S. Department of Agriculture
[5]ISWS: Illinois State Water Survey



[6]UIUC: University of Illinois at Urbana Champaign, USA
* Tile flow data during 2000 for site B was corrupted and was not used in this study.

35





**Table 2** Adjusted parameter values for plant growth simulation, and parameters used for model calibration

| Parameter | Description | Initial value | | Adjusted value | |
|---|---|---|---|---|---|
| | | corn | soybean | corn | soybean |
| Adjusted parameter values for corn and soybean growth simulation | | | | | |
| BIO_E | Radiation-use efficiency ((kg ha$^{-1}$)/(MJ m$^{-2}$)) | 39 | 25 | 36 | 25 |
| PHU | Potential heat units | 1556 | 1556 | 1500 | 1250 |
| T_BASE | Minimum temperature for plant growth (℃) | 8 | 10 | 8 | 8 |
| HVSTI | Harvest index for optimal growing conditions | 0.50 | 0.31 | 0.50 | 0.40 |
| CPYLD | Normal fraction of phosphorus in yield (kg P kg$^{-1}$ yield) | 0.0016 | 0.0091 | 0.0016 | 0.0067 |
| Parameters used for various processes during model calibration | | | | | |
| **Parameter** | **Description** | **Process** | | | |
| ICN | CN method flag: 0 use traditional SWAT method, which bases CN on soil moisture, 1 use method which bases CN on plant ET | Surface runoff | | | |
| CN2 | Soil moisture condition II curve number | | | | |
| CNCOEF | Plant ET curve number coefficient | | | | |
| R2ADJ | Curve number retention parameter adjust factor | | | | |
| SURLAG | Surface runoff lag coefficient | | | | |
| TDRAIN | Time to drain soil to field capacity (h) | Tile drains | | | |
| GDRAIN | Drain tile lag time (h) | | | | |
| DEP_IMP | Depth to impervious layer (mm) | | | | |
| LATKSATF | Multiplication factor to determine lateral saturated hydraulic conductivity | | | | |
| SDRAIN | Tile spacing (mm) | | | | |
| SOL_K(1) | Saturated hydraulic conductivity (mm h$^{-1}$) | | | | |
| ESCO | Soil evaporation compensation factor | Evapotranspiration | | | |
| SFTMP | Snowfall temperature (°C) | Snow | | | |
| SMTMP | Snow melt base temperature (°C) | | | | |
| GW_DELAY | Groundwater delay time (days) | Groundwater | | | |
| RCHRG_DP | Deep aquifer percolation fraction | | | | |
| SOL_AWC(1) | Available water capacity of the soil layer (mm H$_2$0 mm$^{-1}$ soil) | Soil water | | | |
| ADJ_PKR | Peak rate adjustment factor for sediment routing in the subbasin (tributary channels) | Sediment losses | | | |
| SPEXP | Exponent parameter for calculating sediment re-entrained in channel sediment routing | | | | |
| CH_COV1 | Channel erodibility factor | | | | |
| HRU_SLP | Average slope steepness (m m$^{-1}$) | | | | |
| SLSUBBSN | Average slope length (m) | | | | |
| USLE_K | USLE equation soil erodibility (K) factor (0.013 (metric ton m$^2$ h)/(m$^3$ metric ton cm)) | | | | |
| USLE_C | Minimum value of USLE C factor for water erosion | | | | |
| CMN | Rate factor for mineralization for humus active organic nutrients (N) | Nitrate losses | | | |
| RCN | Concentration of nitrogen in rainfall (mg N L$^{-1}$) | | | | |
| NPERCO | Nitrogen concentration reduction coefficient | | | | |
| SDNCO | Denitrification threshold water content | | | | |
| CDN | Denitrification exponential rate coefficient | | | | |



**Table 3** Calibrated values of adjusted parameters for tile flow and nitrate-N calibration of SWAT at sites B, E, Bs, Es and R5

| Parameter | Range | Calibrated value | | | | | | | | | | | |
|---|---|---|---|---|---|---|---|---|---|---|---|---|---|
| | | Site B | | | Site E | | | Site Bs | | Site Es | | Site R5 | |
| | | 528 | 615 | 645 | 528 | 615 | 645 | 615 | 645 | 615 | 645 | 528 | 615 |
| ICN | | 1 | 1 | 0 | 0 | 0 | 0 | 0 | 0 | 0 | 0 | 1 | 0 |
| CN2 | -0.2~-0.1 | 61 | 63 | - | 64 | 65 | - | 60 | - | 60 | - | -0.2 | -0.2 |
| CNCOEF | 0.5~2 | 0.83 | 0.98 | - | - | - | - | - | - | - | - | 0.58 | - |
| R2ADJ | 0~1 | - | - | 0.96 | - | - | 0.97 | - | 0.88 | - | 0.81 | - | - |
| SURLAG | 0.5~2 | 1.91 | 1.62 | 0.97 | 0.61 | 1.59 | 0.73 | 1.78 | 1.80 | 1.82 | 1.83 | 0.77 | 1.03 |
| TDRAIN (h) | 24~48 | 26 | - | - | 25 | - | - | - | - | - | - | 26 | - |
| GDRAIN (h) | 24~48 | 25 | - | - | 26 | - | - | - | - | - | - | 25 | - |
| DEP_IMP (mm) | 1200~3600 | 3400 | 2300 | 2600 | 2100 | 1200 | 1500 | 3400 | 2000 | 3100 | 1900 | 3600 | 2700 |
| LATKSATF | 0.01~4 | - | 2.2 | 1.02 | - | 0.07 | 0.26 | 1.68 | 0.48 | 1.89 | 0.28 | - | 1.05 |
| SDRAIN (mm) | 25000~50000 | - | 33000 | 37000 | - | 28000 | 28000 | 36000 | 29000 | 29000 | 41000 | - | 38000 |
| SOL_K(1) (mm h⁻¹) | -0.8~0.8 | -0.24 | 0.68 | -0.79 | 0.32 | -0.62 | 0.62 | 0.03 | 0.52 | -0.17 | 0.36 | -0.26 | 0.07 |
| ESCO | 0.8~0.99 | 0.94 | 0.95 | 0.86 | 0.82 | 0.88 | 0.91 | 0.85 | 0.85 | 0.90 | 0.95 | 0.93 | 0.98 |
| SFTMP (°C) | -5~5 | -1.79 | 2.77 | -4.47 | -1.99 | 1.34 | 3.35 | -4.96 | 4.37 | 4.53 | 3.97 | 0.58 | -4.25 |
| SMTMP (°C) | -5~5 | -2.28 | 2.59 | -3.78 | 3.39 | 0.86 | -1.52 | -1.4 | 4.8 | 0.11 | 1.57 | 0.99 | 2.08 |
| GW_DELAY (days) | 10~40 | 16 | 29 | 22 | 27 | 21 | 20 | 12 | 16 | 32 | 19 | 37 | 25 |
| RCHRG_DP | 0~0.3 | 0.05 | 0.09 | 0.28 | 0.04 | 0.11 | 0.03 | 0.08 | 0.20 | 0.21 | 0.28 | 0.72 | 0.56 |
| SOL_AWC(1) | -0.2~0.2 | 0.05 | -0.19 | 0.18 | 0.04 | -0.09 | -0.03 | 0.03 | -0.16 | 0.19 | 0.15 | 0.06 | -0.03 |
| ADJ_PKR | 0.5~2 | - | - | - | - | - | - | - | 1.75 | - | 0.74 | 1.10 | 1.16 |
| SPEXP | 1~2 | - | - | - | - | - | - | - | - | - | - | 1.50 | 1.94 |
| CH_COV1 | 0~1 | - | - | - | - | - | - | - | | - | | 0.38 | 0.31 |
| HRU_SLP (m m⁻¹) | 0~0.02 | - | - | - | - | - | - | - | 0 | - | 0.02 | 0.02 | 0 |
| SLSUBBSN (m) | -0.1~0.1 | - | - | - | - | - | - | - | | - | | 0.08 | 0.03 |
| USLE_K(1) | -0.1~0.1 | - | - | - | - | - | - | - | 0.07 | - | 0.1 | 0.1 | -0.06 |
| USLE_C{19} | -0.25~0.25 | - | - | - | - | - | - | - | 0.00 | - | 0.24 | 0.23 | 0.15 |
| USLE_C{56} | -0.25~0.25 | - | - | - | - | - | - | - | -0.17 | - | -0.12 | 0.15 | 0.07 |
| CMN | 0.0003~0.003 | 0.02 | 0.02 | - | 0.0003 | 0.02 | - | - | 0.01 | - | 0.02 | 0.0003 | 0.03 |
| RCN (mg N L⁻¹) | 0~15 | 11 | 15 | - | 10 | 11 | - | - | 6 | - | 5 | 11 | 0.1 |
| NPERCO | 0~1 | 0.84 | 0.01 | - | 0.53 | 0.48 | - | - | 0.99 | - | 1 | 0.99 | 0.99 |
| SDNCO | 0~1.5 | 1.25 | 1.26 | - | 1.02 | 1.39 | - | - | 1.30 | - | 0.93 | 1 | 1.46 |
| CDN | 0~1 | 0.01 | 0.02 | - | 0.33 | 0.28 | - | - | 1 | - | 1 | 0.06 | 0 |

Negative value for CN2, and value for SOL_K(1), SOL_AWC(1) (mm $H_2O$ mm⁻¹ soil), USLE_K(1) (0.013 (metric ton m² h)/(m³ metric ton cm)), USLE_C{19}, and USLE_C{56} is relative change to default value. (1) indicates the first soil layer. {19} and {56} represent corn and soybean, respectively.



**Table 4** Performance evaluation of calibrated and validated results at sites B, E, Bs, Es and R5

| | Annual Crop yield (t ha⁻¹) | | Monthly Tile flow (mm) | | | | | | Monthly NO3-N in tile flow (kg ha⁻¹) | | | |
|---|---|---|---|---|---|---|---|---|---|---|---|---|
| | Cali | Vali | Cali | | | Vali | | | Cali | | Vali | |
| Revision | 615 | 615 | 528 | 615 | 645 | 528 | 615 | 645 | 528 | 615 | 528 | 615 |
| **Site B** | | | | | | | | | | | | |
| $P_{BIAS}$ (%) | 13 | 2 | 3 | 14 | -19 | 4 | 3 | -18 | 8 | 33 | 23 | 18 |
| $R^2$ | 0.99 | 0.92 | 0.73 | 0.49 | 0.72 | 0.80 | 0.69 | 0.64 | 0.65 | 0.37 | 0.67 | 0.78 |
| NSE | 0.91 | 0.91 | 0.71 | 0.54 | 0.66 | 0.80 | 0.68 | 0.58 | 0.66 | 0.22 | 0.63 | 0.77 |
| MSE | 0.77 | 0.76 | 0.60 | 0.54 | 0.53 | 0.67 | 0.53 | 0.48 | 0.59 | 0.43 | 0.55 | 0.64 |
| KGE | 0.75 | 0.89 | 0.85 | 0.70 | 0.75 | 0.86 | 0.78 | 0.73 | 0.68 | 0.48 | 0.71 | 0.78 |
| **Site E** | | | | | | | | | | | | |
| $P_{BIAS}$ (%) | -2 | 5 | -37 | -6 | -17 | -10 | 12 | -2 | -1 | 26 | 36 | 20 |
| $R^2$ | 0.95 | 0.92 | 0.68 | 0.51 | 0.6 | 0.75 | 0.52 | 0.56 | 0.72 | 0.61 | 0.38 | 0.55 |
| NSE | 0.95 | 0.88 | -0.77 | 0.5 | 0.54 | -0.2 | 0.5 | 0.53 | -0.09 | 0.5 | 0.21 | 0.5 |
| MSE | 0.80 | 0.71 | -0.20 | 0.28 | 0.27 | 0.04 | 0.31 | 0.34 | 0.08 | 0.3 | 0.18 | 0.34 |
| KGE | 0.95 | 0.91 | -0.05 | 0.6 | 0.71 | 0.15 | 0.60 | 0.74 | 0.24 | 0.66 | 0.45 | 0.65 |

| | Monthly Surface runoff (mm) | | | | Monthly Sediment (t ha⁻¹) | | Monthly Nitrate in runoff (kg ha⁻¹) | |
|---|---|---|---|---|---|---|---|---|
| | Cali | | Vali | | Cali | Vali | Cali | Vali |
| Revision | 615 | 645 | 615 | 645 | 645 | 645 | 645 | 645 |
| **Site Bs** | | | | | | | | |
| $P_{BIAS}$ (%) | -614 | -26 | -475 | -74 | -5 | 37 | 79 | 53 |
| $R^2$ | 0.23 | 0.88 | 0.76 | 0.80 | 0.96 | 0.13 | 0.14 | 0.01 |
| NSE | -4.7 | 0.81 | -5.95 | 0.56 | 0.95 | 0.11 | 0.06 | -0.32 |
| MSE | -2.36 | 0.48 | -1.70 | 0.41 | 0.74 | 0.48 | 0.43 | 0.29 |
| KGE | -5.33 | 0.58 | -4.22 | 0.18 | 0.86 | 0.10 | -0.34 | -0.11 |
| **Site Es** | | | | | | | | |
| $P_{BIAS}$ (%) | -107 | -18 | -143 | -99 | 32 | 22 | 25 | 83 |
| $R^2$ | 0.71 | 0.82 | 0.55 | 0.48 | 0.79 | 0.11 | 0.33 | 0.005 |
| NSE | 0.50 | 0.82 | -0.85 | -0.28 | 0.46 | 0.08 | 0.27 | -0.07 |
| MSE | 0.28 | 0.57 | 0.01 | 0.15 | 0.39 | 0.27 | 0.31 | 0.35 |
| KGE | -0.10 | 0.78 | -0.67 | -0.15 | 0.24 | 0.12 | 0.17 | -0.54 |

| | Monthly Flow (cms) | | | | Monthly Sediment (t) | | | | Monthly Nitrate (kg) | | | |
|---|---|---|---|---|---|---|---|---|---|---|---|---|
| | Cali | | Vali | | Cali | | Vali | | Cali | | Vali | |
| Revision | 528 | 615 | 528 | 615 | 528 | 615 | 528 | 615 | 528 | 615 | 528 | 615 |
| **Site R5** | | | | | | | | | | | | |
| $P_{BIAS}$ (%) | -36 | -48 | -1 | -11 | 62 | 10 | -141 | -474 | 11 | 17 | 31 | 37 |
| $R^2$ | 0.85 | 0.84 | 0.68 | 0.63 | 0.27 | 0.45 | 0.31 | 0.76 | 0.56 | 0.63 | 0.70 | 0.67 |
| NSE | 0.77 | 0.73 | 0.60 | 0.48 | 0.18 | 0.45 | -1.05 | -9.67 | 0.33 | 0.61 | 0.57 | 0.58 |
| MSE | 0.50 | 0.43 | 0.36 | 0.26 | 0.40 | 0.46 | 0.07 | -1.77 | 0.40 | 0.50 | 0.51 | 0.50 |
| KGE | 0.63 | 0.50 | 0.80 | 0.71 | 0.001 | 0.56 | -0.63 | -4.61 | 0.65 | 0.71 | 0.64 | 0.48 |

Cali and Vali represent calibration and validation, respectively.



**Fig. 1**

![Map of study area showing river, subsurface, and surface stations across Champaign Co., Vermilion Co., Douglas Co., and Edgar Co. in Illinois, with a 5 km scale bar and north arrow. Station labels include R5, Bs, B, Es, E, and Dam.]





**Fig. 2**





**Fig. 3**



**Fig. 4**





**Fig. 5**





**Fig. 6**

