# Peer review of "Comparison of performance of tile drainage routines in SWAT 2009 and 2012 in an extensively tile-drained watershed in the Midwest"

_Hydrology and Earth System Sciences, 2017_

## Short Comment (SC1) · 19 Feb 2017

Title: Comparison of performance of tile drainage routines in SWAT 2009 and 2012 in an extensively tile-drained watershed in the Midwest

Authors: Tian Guo, Margaret Gitau, Venkatesh Merwade, Jeffrey Arnold, Raghavan Srinivasan, Michael Hirschi, and Bernard Engel Journal: Hydrology and Earth System Sciences

Review:

In this study the old tile drainage routine in SWAT2009 (Rev.528) and the new tile drainage routine in SWAT2012 (Rev.615 and Rev.645) are used in the simulations to

evaluate the performance of both tile drainage routines.

Based on this review, the following comments are made:

1) From the reader's point of view, the current version of the manuscript does not have a scientific merit. This manuscript is yet another research work from SWAT community on calibration, validation, and application of SWAT. The following questions are raised:

a) Does this manuscript develop/devise a new methodology? b) Does this manuscript develop/devise a new tool? c) Does this manuscript develop/propose a new theory?

2) From the reader's point of view, it is hard to understand the motivation of this paper. As per the current version of the paper, referring to line number 27 on page number 18, in this study the old tile drainage routine in SWAT2009 (Rev.528) and the new tile drainage routine in SWAT2012 (Rev.615 and Rev.645) were used in the simulations to evaluate the performance of both tile drainage routines. The following questions are raised:

a) Did the developers of SWAT released a revision (645 or 615) without evaluating the model outcome? b) Did the developers of SWAT released a revision (645 or 615) with an anticipation of getting poor model outcome? c) Why did the developers include new routines (Rev.615 and Rev.645)?

3) In the current version of the paper, the authors state that SWAT2012 revision 645, which "improved" the soil moisture based curve number calculation method, has not been fully "tested". Why did the developers improved the soil moisture based curve number calculation method? Was it to get poor model outcome? Did the developers improve the method without testing?

4) From the reader's point of view, the introduction of the manuscript needs to re-written. In the current version of the manuscript, the introduction is built with many equations. From the reader's point of view, a section with all these equations need to be introduced after the introduction. This will help the authors to have an introduction

to highlight the need of the research.

5) From the reader's point of view, some of the paragraphs in the introduction are not coherent.

6) From the reader's point of view, the conclusions need to be re-written. Some of the words (e.g., site B, site E, and R5) in the current version of the paper need to be deleted. The actual locations of the sites need to be mentioned in the conclusion.

7) In the abstract, the authors claim that both the routines provided reasonable but unsatisfactory uncalibrated flow and nitrate loss results. The authors should clearly state the meaning of "reasonable but unsatisfactory". Moreover, the authors need to state the temporal scale of their statement.

8) In the abstract, the authors claim that the new routine provided acceptable simulated tile flow and nitrate in tile flow for both field sites with random pattern tile and constant tile spacing. However, in the current version of the paper, the reader is unable to find more detail about the random pattern. Moreover, it would be more meaningful if the authors relate these patterns to the adopted equations shown in equations (3-5).

9) In the current version of the paper, it is understood that there exists a coefficient named "drainage coefficient" (DC in equation-5) in SWAT 2009 and SWAT 2012. The authors also state that a coefficient named "drainage coefficient "(DRAIN_CO) was included in the new tile drainage routine in SWAT2012. Does SWAT2012 in its tile drainage routine have two drainage coefficients?

10) The authors need to clearly state the difference between SWAT2012 Rev.615 and SWAT2012 Rev.645.

11) As per the current version of the paper, a coefficient named drainage coefficient (DRAIN_CO) was included in the new tile drainage routine in SWAT2012 to "control "peak drain flow. However, in the current version of the paper, the old tile drainage routine in SWAT2009 (Rev.528) and the new tile drainage routine in SWAT2012 (Rev.615

and Rev.645) were used to simulate monthly tile flow, nitrate in tile flow, surface runoff, and sediment and nitrate in surface runoff at field sites, and monthly flow, sediment and nitrate in flow at a river station. Therefore, it is unclear about the motivation of this research work. Moreover, it would be meaningful if the authors show the equation that uses DRAIN_CO.

12) The Fig 1 needs to be checked by a GIS professional. From the reader's point of view, the Fig 1 is meaningless. Moreover, there is an asterisk within the IL boundary. This asterisk should be related to the main figure. The abbreviation "Co." is not understood. The caption of the figure needs to be self-illustrative. The county borders also need to be checked. Do they intersect orthogonally?

13) In Fig 1, is the river station R5 shared by both the counties (i.e., Vermillion and Champaign counties)?

14) The authors need to state few lines about the methodology used to get the drainage areas of subsurface stations and surface runoff stations.

15) As per the current version of the paper(line number five on page number six), daily nitrate and sediment load was computed by multiplying water discharges with nitrate concentration (Yuan et al., 2000). How did the authors compute the daily sediment load?

16) As per the current version of the paper (line number eight on page number six), nitrate and sediment loads were computed by multiplying the concentration at a specific time by half the flow volume since the last concentration measurement plus half the flow volume from the concentration measurement to the next concentration measurement (Kalita et al., 2006; Yuan et al., 2000).The authors also state that nitrate and sediment concentration data were not available for "every day" that water discharge occurred. Therefore, the adopted methodology is not understood. Do the authors have nitrate and sediment concentration data every two days?

http://research.abzwater.com/review/ABZR7.pdf

---

## Referee Comment (RC1) · Anonymous Referee #1 · 14 Mar 2017

This paper aims to evaluate the performance of new physically based tile drainage routines proposed by Hooghoudt and Kirkham. The study is conducted in a small watershed (518 km2) in the Midwest USA. The main objective is to compare simulated flow, tile flow, runoff, nitrate in tile flow and sediment load results for the new tile drainage routines in SWAT2012 and the old one in SWAT2 009 in the LVR watershed and determine which routine provides a better model fit with observed values. Testing of the new routines and identification of parameter sets is given as the primary motivation for this research. In my opinion, the given motivation and objective add very little to the scientific knowledge, thus, do not merit publication in HESS Journal in the current form. The authors claim that the parameter set obtained from this study provide guidance for field and watershed level applications. In fact, this is not a new and significant finding. Moreover, author do not provide any discussion on physical basis of the selected parameters. Neither differences due to spatial scales are mentioned. Some of the parameter values are also hard to understand, for instance, the range of snow fall and snow melt parameters seems too large (-5 to 5 oC). From physical process point of view, it is hard to explain why these parameters are so different in such a small and mildly sloped watershed? To mention another example, why fitting values of SURLAG differ between sites (how scaling in hydrology may guide explaining this?). Similar can be said for other parameters like curve number, sediment and nitrogen related parameters. Therefore, the currently presented parameter sets adds very little to the available knowledge. A critical discussion on the fitted parameter values, at least explaining physical process related reasons and issues of spatial scales, is recommended. Another major problem is difficulty in following the structure of the paper. Presentation of calibration and validation results for each site demonstrates lot of repetition. This obstruct clarity and the readers could soon start feeling bored as same information comes again without any new insights and deeper discussion. One way of rectifying this issue could be by fully restructuring the paper. For example, results can be separately presented for each indicator (crop yields, flows, sediment, and nitrate) rather than per site. This can also facilitate physical explanation and scale issues when results of all sites for one indicator are combined together. For instance, when it comes to peak flow or runoff simulations, one can see where it was simulated well, at R5 or B or E etc, and then what could be the governing factors (geography, tile drainage density, variation in hydraulic conductivity, effect of CN etc). Although the study mentions previous research on testing the new tile drainage routine, the results of this study are not compared with the previous findings. A detailed comparison with the previous studies would help to understand and position this work much better. While doing so, the authors should at least include topics related to parametrization, characteristics of the studied watersheds, performance evaluation results. Additionally, some very useful comments are made by S. Mylevaganam. In general, I see them valid and constructive

(though critical) and could be helpful for improving the manuscript.

---

## Referee Comment (RC2) · Anonymous Referee #2 · 11 Apr 2017

The author evaluates performance of tile drainage routines in SWAT 2009 (revision 528) and 2012 (revisions 615 and 645) at two points in mildly sloped LVR watershed based on runoff, Nitrate, etc., I suggest major revision owing to following comments below: i)I think scientific merit of this paper can be improved from its current form by showing how (under changing climate and irrigation practices) contamination of water has changed owing to tile drainage; after setting up well calibrated routines and simulating N-contamination for long-term till last year or so. ii) Author can try to discuss on how modified curve number improves SWAT 2012 tile drainage routines. iii) Fig 3c and d, Tile flow simulated from Rev.528 show constant overestimation at E and hence I feel still there is scope of improving (calibration) parameters. This may be leading to

following conclusion on page 19 line4-5: old routine were better at site B, while new routine were better as site E. Difference in performance of different routines at B and E should be discussed. Is this based on different routines performing differently in different land-use at B and E or is there other physical process of routines linked to this difference. iv) The area covered by surface and sub-surface station is as low as in range of 0.05 km2. What is HRU size corresponding to drainage area for B and E? This information will reveal how well drainage is simulated in the considered drainage area. v) Leave-few-year out approach may be more suitable for calibration and validation. vi) Introduction can be reconstructed. In current form science question are repeated at two places on page 2 line 5 and page 4 line 32. vii) (line 20) Explanation is required on how uncalibrated routines give 'reasonable but unsatisfactory' performance. viii) Page 5 line 25 citation is improper ix) Page 9 line 27 variables of equation are not properly defined. x) Repetition: Page 14 line 13-14, Two sentences can be merge in 1. Page 14-19 looks like repetition of sentences. xi) Page 18 line 31, 'both routines' which two? Is not clear.

---

## Short Comment (SC2) · 22 Apr 2017

Comments from Referee # 2: I think scientific merit of this paper can be improved from its current form by showing how (under changing climate and irrigation practices) contamination of water has changed owing to tile drainage; after setting up well calibrated routines and simulating N-contamination for long-term till last year or so.

R: We thank the referee # 2 for valuable suggestions to our manuscript. We agree that the scientific merit of our manuscript need to be well described. We have discussed in the Introduction section, "Subsurface tile drainage systems could move out of the soil surface and convey soluble nitrate-N from the crop root zone. Nitrate coming from tile drains has been considered the main source of nitrate in rivers and streams in the

[Figure]

Midwestern US. Additionally, 89 % - 95 % of nitrate losses in a ditch catchment were transported by the tile drainage system of the catchment." (page 2 line 24-27). We will incorporate more discussion about impacts of tile drainage systems on water quality such as nitrate losses under changing climate, especially under changing precipitation across years.

The research results in this manuscript could provide guidance for selection of tile drainage routines and related parameter sets for tile drainage simulation at both field and watershed scales. For example, well calibrated routines and related parameter sets in this study have been used for modeling of the impacts of bioenergy crop scenarios on streamflow, tile flow, sediment and nitrate losses in the LVR watershed from 1990 to 2008 (Guo et al., 2017, unpublished).

ii) Author can try to discuss on how modified curve number improves SWAT 2012 tile drainage routines. R: Yes, the newly added curve number calculation retention parameter adjustment factor in Rev.645 calculates curve numbers reasonably well based on the soil moisture retention curve from field capacity to saturation, and can partition surface runoff and tile flow well. Thus, the modified curve number improve surface runoff simulation and this improve tile drainage simulation in SWAT 2012 (Figs. 2c and 3c).

iii) Fig 3c and d, Tile flow simulated from Rev.528 show constant overestimation at E and hence I feel still there is scope of improving (calibration) parameters. This may be leading to following conclusion on page 19 line4-5: old routine were better at site B, while new routine were better as site E. Difference in performance of different routines at B and E should be discussed. Is this based on different routines performing differently in different land-use at B and E or is there other physical process of routines linked to this difference. R: We thank the referee # 2 for this thought-provoking suggestion. Sites B and E have similar land use, corn and soybean, but with different rotations. Difference in performance of different routines at B and E may be mainly caused by different climatic characteristics of two sites, and physical process in the old routines. The old routine in Rev.528 has the potential to overestimate tile flow peaks,

since simulated tile flow by the old routine was controlled by a simple drawdown time parameter (TDRIAN), and tiles were allowed to carry an unlimited maximum of water no matter how intense the rainfall. Moreover, when water table was lower than tiles, the old routine could not calculate tile flow. Thus, Rev.528 has the potential to under-estimate tile flow during dry periods. Thus, Rev.528 could not simulate tile flow peaks and tile flow during dry periods at stie E. As I discussed on page 12 line 28, "However, Rev.528 simulated tile flow was overestimated at tile flow peaks in November 1992, May 1996, March 1997 (Fig. 3c), May and June of 1998, December 2001, and Febru-ary, April and May of 2002 (Fig. 3d). Rev.528 simulated tile flows were underestimated from May to October in 1992, from June to November in 1994, from July in 1995 to March in 1996 (Fig. 3c), from May in 1999 to February in 2000, from May to August in 2001, and from July to December in 2002 (Fig. 3d).".

iv) The area covered by surface and sub-surface station is as low as in range of 0.05 km2. What is HRU size corresponding to drainage area for B and E? This information will reveal how well drainage is simulated in the considered drainage area. R: Yes, we have the same concern. HRU size in SWAT is 14.18 and 0.72 km2, respectively. HRU size in SWAT is larger than the size of station, which could not represent the size of each individual tile. As we mentioned in the Limitation section, there is an opportunity to improve the representation of tile drainage systems in SWAT, especially for individual tiles. We believe that better representation of size and spatial information of tile drainage systems can improve simulation of tile drainage.

v) Leave-few-year out approach may be more suitable for calibration and validation. R: Does leave-few-year out approach mean leaving out few year, such as one year monthly observed data, to use as the validation data, and using the remaining ob-served data for calibration? If so, this approach will be suitable for our study, and we will not expect obvious differences between statistics for model evaluation for the ap-proach used in this study (7 and 6 years of monthly data for calibration and validation, respectively) and for leave-few-year out approach.

vi) Introduction can be reconstructed. In current form science question are repeated at two places on page 2 line 5 and page 4 line 32. R: We thank the referee # 2 for the detailed suggestions to the structure and description of this manuscript. We are very grateful about it. The Science question on page 2 line 5 has been removed. The introduction will be reorganized to improve the flow of the manuscript.

vii) (line 20) Explanation is required on how uncalibrated routines give 'reasonable but unsatisfactory' performance. R: "Both routines provided reasonable but unsatisfactory uncalibrated flow and nitrate loss results." has been changed to "Both routines provided reasonable but unsatisfactory (NSE < 0.5) uncalibrated flow and nitrate loss results for a mildly-sloped watershed with low runoff."

viii) Page 5 line 25 citation is improper R: Citation on page 5 line 25 has been corrected.

ix) Page 9 line 27 variables of equation are not properly defined. R: I have change "Where Obs and Sim represent observed and simulated data, respectively." to "Where Obs and Sim represent the ith observed and simulated monthly data, respectively. And n is the total number of months. (Obs) ÌĚ and (Sim) ÌĚ represent the average values of the observed and simulated monthly data, respectively."

x) Repetition: Page 14 line 13-14, Two sentences can be merge in 1. Page 14-19 looks like repetition of sentences. R: The sentence on page 14 line 13-14 has been condensed to "Performance of the modelled monthly surface runoff from Rev.645 at site Bs during calibration and validation was satisfactory from Rev.645 and unsatisfactory from Rev.615.".

The sentences from page 14 to 19 will be reorganized to avoid repetition.

xi) Page 18 line 31, 'both routines' which two? Is not clear. R: Both routines represented the old tile drainage routine in SWAT2009 (Rev.528) and the new tile drainage routine in SWAT2012 (Rev.615 and Rev.645), which was mentioned in the last sentence. I have changed 'both routines' to both the old and new routines'.

[Figure]

Guo, T., Raj, C., Chaubey, I., Gitau, M., Arnold, J. G., Srinivasan, R., Kiniry, J. R. & Engel, B. A. (2017). Evaluation of bioenergy crop growth and the impacts of bioenergy crops on streamflow, tile drain flow and nutrient losses in an extensively tile-drained watershed using SWAT (under review).

---

## Short Comment (SC3) · 22 Apr 2017

Comments from Referee # 1ïïjŽTesting of the new routines and identification of parameter sets is given as the primary motivation for this research. In my opinion, the given motivation and objective add very little to the scientific knowledge, thus, do not merit publication in HESS Journal in the current form. The authors claim that the parameter set obtained from this study provide guidance for field and watershed level applications. In fact, this is not a new and significant finding. Moreover, author do not provide any discussion on physical basis of the selected parameters.

R: We thank the referee #1 for the suggestions to our manuscript. Yes, we will provide more discussion on physical basis of the calibrated parameters and describe the

relationship between the parameters and physical process of tile drainage. But we do not agree that our manuscript add very little to the scientific knowledge, or this is not a new and significant finding. We agree that tile drainage modeling using SWAT has been conducted in other watersheds. But this is the first one conducted in the LVR watershed. The soil and climatic characteristics, tile drainage system pattern, and management practices vary in different watershed. The research results in this manuscript could provide guidance for selection of tile drainage routines and related parameter sets for tile drainage simulation at both field and watershed scales. For example, well calibrated routines and related parameter sets in this study have been used for modeling of the impacts of bioenergy crop scenarios on streamflow, tile flow, sediment and nitrate losses in the LVR watershed from 1990 to 2008 (Guo et al., 2017, unpublished). Thus, this study is innovative and important.

Comments from Referee # 1ïïjŽSome of the parameter values are also hard to understand, for instance, the range of snow fall and snow melt parameters seems too large (-5 to 5 °C). From physical process point of view, it is hard to explain why these parameters are so different in such a small and mildly sloped watershed? To mention another example, why fitting values of SURLAG differ between sites (how scaling in hydrology may guide explaining this?). Similar can be said for other parameters like curve number, sediment and nitrogen related parameters. Therefore, the currently presented parameter sets adds very little to the available knowledge. A critical discussion on the fitted parameter values, at least explaining physical process related reasons and issues of spatial scales, is recommended.

R: Yes, we agree that the range of snow fall and melt parameters are large. We will narrow the range for the selected parameter and improve our calibration. Land use, soil, climate, pattern of tile drainage systems, and management practices are different in different stations, thus it is reasonable to have different calibrated parameter sets for tile drainage simulation. But they are similar with each other at different stations, rather than so different. We would like to thank the referee # 1 for the suggestions

about a critical discussion on the fitted parameters. We will incorporate more in-depth discussion about how the calibrated parameter sets for different routines represent the physical process of tile drainage.

Comments from Referee # 1ïïjŽAnother major problem is difficulty in following the structure of the paper. Presentation of calibration and validation results for each site demonstrates lot of repetition. This obstruct clarity and the readers could soon start feeling bored as same information comes again without any new insights and deeper discussion. One way of rectifying this issue could be by fully restructuring the paper. For example, results can be separately presented for each indicator (crop yields, flows, sediment, and nitrate) rather than per site. This can also facilitate physical explanation and scale issues when results of all sites for one indicator are combined together. For instance, when it comes to peak flow or runoff simulations, one can see where it was simulated well, at R5 or B or E etc, and then what could be the governing factors (geography, tile drainage density, variation in hydraulic conductivity, effect of CN etc).

R: We thank the referee #1 for the constructive suggestions about improving the structure of the manuscript. We will reorganize the results and discussion and present results for each indicator, to avoid repetition and improve flow of the manuscript. We will also relate parameter sets with physical process of tile drainage, and compare the performance of the old and new routines in simulating the same indicator at different sites. Yes, the same routine had different performance at different sites, which was mainly caused by different climatic characteristics and how the routine simulates tile flow. For example, old routine was better at site B, while new routine was better as site E. Difference in performance of different routines at B and E may be mainly caused by different climatic characteristics of two sites, and physical process in the old routines. The old routine in Rev.528 has the potential to overestimate tile flow peaks, since simulated tile flow by the old routine was controlled by a simple drawdown time parameter (TDRIAN), and tiles were allowed to carry an unlimited maximum of water no matter how intense the rainfall. Moreover, when water table was lower than tiles, the old routine could not calculate tile flow. Thus, Rev.528 has the potential to underestimate tile flow during dry periods. Thus, Rev.528 could not simulate tile flow peaks and tile flow during dry periods at site E.

Comments from Referee # 1ïijŽAlthough the study mentions previous research on testing the new tile drainage routine, the results of this study are not compared with the previous findings. A detailed comparison with the previous studies would help to understand and position this work much better. While doing so, the authors should at least include topics related to parametrization, characteristics of the studied watersheds, performance evaluation results.

R: We thank the referee # 1 for this valuable suggestion. The previous study on testing the new tile drainage routine evaluated performance of the routine in simulating streamflow on a tile drained watershed, without observed tile flow data at field scales. Thus, we will compare the previous studies with our calibration and validation at site R5, to improve the understanding of this study. The calibrated parameter sets, difference between characteristics of watersheds, and routine performance will also be incorporated.

Guo, T., Raj, C., Chaubey, I., Gitau, M., Arnold, J. G., Srinivasan, R., Kiniry, J. R. & Engel, B. A. (2017). Evaluation of bioenergy crop growth and the impacts of bioenergy crops on streamflow, tile drain flow and nutrient losses in an extensively tile-drained watershed using SWAT (under review).

---

## Author Comment (AC1) · 18 May 2017

Interactive comments # 1ïïjŽ Based on this review, the following comments are made: 1) From the reader's point of view, the current version of the manuscript does not have a scientific merit. This manuscript is yet another research work from SWAT community on calibration, validation, and application of SWAT. The following questions are raised: a) Does this manuscript develop/devise a new methodology? b) Does this manuscript develop/devise a new tool?

c) Does this manuscript develop/propose a new theory? R: We thank Dr. Mylevaganam for the detailed and valuable suggestions to our manuscript. We agree that the scientific merit of our manuscript needs to be well described. However, we do not agree that the manuscript does not have scientific metric. The SWAT calibration, validation and application research has scientific metric. We have discussed the importance in the Introduction section, "Subsurface tile drainage systems could move out of the soil surface and convey soluble nitrate-N from the crop root zone. Nitrate coming from tile drains has been considered the main source of nitrate in rivers and streams in the Midwestern US. Additionally, 89 % - 95 % of nitrate losses in a ditch catchment were transported by the tile drainage system of the catchment." (page 2 line 24-27). Moreover, the research results in this manuscript could provide guidance for selection of tile drainage routines and related parameter sets for tile drainage simulation at both field and watershed scales. For example, well calibrated routines and related parameter sets in this study have been used for modeling of the impacts of bioenergy crop scenarios on streamflow, tile flow, sediment and nitrate losses in the LVR watershed from 1990 to 2008 (Guo et al., 2017, unpublished). Thus, this study is innovative and important.

Interactive comments # 1ïijŽ 2) From the reader's point of view, it is hard to understand the motivation of this paper. As per the current version of the paper, referring to line number 27 on page number 18, in this study the old tile drainage routine in SWAT2009 (Rev.528) and the new tile drainage routine in SWAT2012 (Rev.615 and Rev.645) were used in the simulations to evaluate the performance of both tile drainage routines. The following questions are raised: a) Did the developers of SWAT released a revision (645 or 615) without evaluating the model outcome? b) Did the developers of SWAT released a revision (645 or 615) with an anticipation of getting poor model outcome? c) Why did the developers include new routines (Rev.615 and Rev.645)? R: Rev. 615 has been extensively tested in previous studies and provided satisfactory outcomes. However, curve number needs to be reduced tremendously to obtain acceptable flow results in mildly-sloped areas. This is the first manuscript to test Rev. 645. We did not

expect poor model outcomes from revision 645, which provided satisfactory surface runoff, and sediment and nitrate in surface runoff results at site Bs and Es. Rev.615 incorporates a new tile drainage routine (from DRAINMOD) to better represent physical process of tile drainage, and Rev.645 incorporates an improved curve number calculation method to improve surface runoff simulation in mildly sloped areas. That is why we included these two versions.

Interactive comments # 1ïijŽ 3) In the current version of the paper, the authors state that SWAT2012 revision 645, which "improved" the soil moisture based curve number calculation method, has not been fully "tested". Why did the developers improved the soil moisture based curve number calculation method? Was it to get poor model outcome? Did the developers improve the method without testing? R: The original soil moisture based curve number calculation method did not reasonably simulate surface runoff, unless curve number was reduced significantly for mildly sloped areas. This manuscript tested the modified curve number calculation method.

Interactive comments # 1ïijŽ 4) From the reader's point of view, the introduction of the manuscript needs to rewritten. In the current version of the manuscript, the introduction is built with many equations. From the reader's point of view, a section with all these equations need to be introduced after the introduction. This will help the authors to have an introduction to highlight the need of the research. R: We thank Dr. Mylevaganam for valuable suggestions to our manuscript. The introduction can be reorganized and focus on the importance of the research, and the equations can be mentioned after the introduction.

Interactive comments # 1ïijŽ 5) From the reader's point of view, some of the paragraphs in the introduction are not coherent. R: We kindly ask Dr. Mylevaganam which paragraphs are not coherent in the introduction. We can remove/condense them.

Interactive comments # 1ïijŽ 6) From the reader's point of view, the conclusions need to be re-written. Some of the words (e.g., site B, site E, and R5) in the current version of

the paper need to be deleted. The actual locations of the sites need to be mentioned in the conclusion. R: We thank Dr. Mylevaganam for the constructive suggestions about improving the structure of the manuscript. We will reorganize the results and discussion and remove the information of site number, to improve flow of the manuscript.

Interactive comments # 1ïïjŽ 7) In the abstract, the authors claim that both the routines provided reasonable but unsatisfactory uncalibrated flow and nitrate loss results. The authors should clearly state the meaning of "reasonable but unsatisfactory". Moreover, the authors need to state the temporal scale of their statement. R: "Both routines provided reasonable but unsatisfactory uncalibrated flow and nitrate loss results." has been changed to "Both routines provided reasonable but unsatisfactory (NSE < 0.5) uncalibrated flow and nitrate loss results for a mildly-sloped watershed with low runoff."

Interactive comments # 1ïïjŽ 8) In the abstract, the authors claim that the new routine provided acceptable simulated tile flow and nitrate in tile flow for both field sites with random pattern tile and constant tile spacing. However, in the current version of the paper, the reader is unable to find more detail about the random pattern. Moreover, it would be more meaningful if the authors relate these patterns to the adopted equations shown in equations (3-5). R: The selected sites incorporated both random pattern tile and constant tile spacing. However, random tile spacing is still represented as a constant tile spacing in the model currently. As we mentioned in the Limitation section, there is an opportunity to improve the representation of tile drainage systems in SWAT, especially for individual tiles. We believe that better representation of size and spatial information of tile drainage systems can improve simulation of tile drainage.

Interactive comments # 1ïïjŽ 9) In the current version of the paper, it is understood that there exists a coefficient named "drainage coefficient" (DC in equation-5) in SWAT 2009 and SWAT 2012. The authors also state that a coefficient named "drainage coefficient "(DRAIN_CO) was included in the new tile drainage routine in SWAT2012. Does SWAT2012 in its tile drainage routine have two drainage coefficients? R: No, DC and DRAIN_CO are the same. We have improved the description to be consistent.

Interactive comments # 1ïijŽ 10) The authors need to clearly state the difference between SWAT2012 Rev.615 and SWAT2012 Rev.645. R: Compared to SWAT2012 Rev.615, SWAT2012 Rev.645 incorporated a new retention parameter adjustment factor (R2ADJ) to modify the soil moisture retention parameter calculation method. R2ADJ was used to modify shape coefficients, and curve number was calculated from capacity to saturation. This method is more reasonable than decreasing curve number directly (page 6, line 29).

Interactive comments # 1ïijŽ 11) As per the current version of the paper, a coefficient named drainage coefficient (DRAIN_CO) was included in the new tile drainage routine in SWAT2012 to "control "peak drain flow. However, in the current version of the paper, the old tile drainage routine in SWAT2009 (Rev.528) and the new tile drainage routine in SWAT2012 (Rev.615 and Rev.645) were used to simulate monthly tile flow, nitrate in tile flow, surface runoff, and sediment and nitrate in surface runoff at field sites, and monthly flow, sediment and nitrate in flow at a river station. Therefore, it is unclear about the motivation of this research work. Moreover, it would be meaningful if the authors show the equation that uses DRAIN_CO. R: The old routine in Rev.528 has the potential to overestimate tile flow peaks, since simulated tile flow by the old routine was controlled by a simple drawdown time parameter (TDRIAN), and tiles were allowed to carry an unlimited maximum of water. Thus, Rev.528 has the potential to overestimate tile flow peaks and nitrate in tile flow at site E. On the contrary, simulated tile flow peaks and nitrate in tile flow peaks from the new routine in Rev. 615 and Rev. 645 captured the observed values fairly well. The motivation of this research is to compare the performance of different tile drainage routines in simulating water quantity and quality at field and watershed scales, and determine the most suitable model for further simulation in the extensively tile-drained watershed. The drainage coefficient represents the maximum amount of water that can be drained from tiles. The tile flow is set equal to the drainage coefficient once tile flow is greater than drainage coefficient (See page 4, line 9).

Interactive comments # 1ïijŽ 12) The Fig 1 needs to be checked by a GIS professional. From the reader's point of view, the Fig 1 is meaningless. Moreover, there is an asterisk within the IL boundary. This asterisk should be related to the main figure. The abbreviation "Co." is not understood. The caption of the figure needs to be self-illustrative. The county borders also need to be checked. Do they intersect orthogonally? R: We have improved Fig.1 to better present study area information.

Interactive comments # 1ïijŽ 13) In Fig 1, is the river station R5 shared by both the counties (i.e., Vermillion and Champaign counties)? R: Yes, the river station R5 is on the county line and shared by both counties.

14) The authors need to state few lines about the methodology used to get the drainage areas of subsurface stations and surface runoff stations. R: We will consultant with data provider and add more information about how the drainage areas were determined.

Interactive comments # 1ïijŽ 15) As per the current version of the paper (line number five on page number six), daily nitrate and sediment load was computed by multiplying water discharges with nitrate concentration (Yuan et al., 2000). How did the authors compute the daily sediment load? R: "daily nitrate and sediment load was computed by multiplying water discharges with nitrate concentration" has been changed to "daily nitrate and sediment load was computed by multiplying water discharges with nitrate and sediment concentration, respectively.

Interactive comments # 1ïijŽ 16) As per the current version of the paper (line number eight on page number six), nitrate and sediment loads were computed by multiplying the concentration at a specific time by half the flow volume since the last concentration measurement plus half the flow volume from the concentration measurement to the next concentration measurement (Kalita et al., 2006; Yuan et al., 2000). The authors also state that nitrate and sediment concentration data were not available for "every day" that water discharge occurred. Therefore, the adopted methodology is not understood. Do the authors have nitrate and sediment concentration data every two days?

[Figure]

R: No, we do not have nitrate or sediment concentration data every two days. Nitrate and sediment concentration data collection was sparsely distributed. Sometimes there were concentration data for several continuous days, but sometimes there were no concentration data for a week. Generally, the nitrate and sediment concentration data were collected twice each month during the study period.

Guo, T., Raj, C., Chaubey, I., Gitau, M., Arnold, J. G., Srinivasan, R., Kiniry, J. R. & Engel, B. A. (2017). Evaluation of bioenergy crop growth and the impacts of bioenergy crops on streamflow, tile drain flow and nutrient losses in an extensively tile-drained watershed using SWAT (under review).

---

## Author Response (AR1)

Responses to reviewers' comments

We thank referees #1 and #2 and interactive reviewer for the valuable suggestions to our manuscript. We have made major revisions in response to reviewers' comments. We recalibrated the models and rewrote the manuscript. We believe the manuscript has been greatly improved as a result of these changes. The pages and lines mentioned in the responses are related to the clean manuscript version.

**Comments from Referee # 1:** This paper aims to evaluate the performance of new physically based tile drainage routines proposed by Hooghoudt and Kirkham. The study is conducted in a small watershed (518 km$^2$) in the Midwest USA. The main objective is to compare simulated flow, tile flow, runoff, nitrate in tile flow and sediment load results for the new tile drainage routines in SWAT2012 and the old one in SWAT2009 in the LVR watershed and determine which routine provides a better model fit with observed values. Testing of the new routines and identification of parameter sets is given as the primary motivation for this research. In my opinion, the given motivation and objective add very little to the scientific knowledge, thus, do not merit publication in HESS Journal in the current form. The authors claim that the parameter set obtained from this study provide guidance for field and watershed level applications. In fact, this is not a new and significant finding. Moreover, author do not provide any discussion on physical basis of the selected parameters.

*Response*: We thank referee #1 for the suggestions to our manuscript. We incorporated discussion on the relationship between the calibrated parameters and physical process of tile drainage system, and sediment and nitrate transport in it. We also improved description of the importance of this study in introduction.

However, we do not agree that our manuscript adds very little to scientific knowledge, or this is not a new and significant finding. We agree that tile drainage modeling using SWAT has been conducted in other watersheds. However, few tile drainage modeling efforts using SWAT have been completed at both field and watershed scales, especially at fields with long term (about 12 years in this study) observed monthly tile flow, nitrate in tile flow, surface runoff, and sediment and nitrate in surface runoff data. Few previous studies have been simulated dynamics between monthly results. Moreover, the soil and weather characteristics, tile drainage system pattern, and management practices vary in different watersheds. This is the first one conducted in the LVR watershed, a typical extensively tiled watershed in the Midwest. In prior studies, RZWQM and DRAINMOD have been calibrated for crop yields, tile flow and nitrate concentration in tile flow at subsurface station, but only for five years of observed data (Singh et al., 2001a; Singh et al., 2001b). In this study, we calibrated tile drainage parameters at paired subsurface and surface stations with different tile spacing (constant tile spacing and random pattern tiles), soil characteristics, different cropping systems (reduced tillage and non-tillage), and different crop management practices. Additionally, besides calibration/validation for hydrology and water quality, we also incorporated calibration and validation for annual biomass yields of corn and soybeans, to make sure the calibrated model can accurately simulate crop growth as well.

This study is innovative and important. It is important and necessary to select an appropriate tile drainage routine suitable for modelling mildly-sloped watersheds in the Midwest with subsurface drainage systems, to accurately simulate subsurface drainage and nutrient and sediment losses, which can benefit research on reducing nitrate from agricultural watershed in the Midwest and alleviating hypoxia in the Gulf of Mexico (Jaynes and James, 2007; Kalita et al., 2007; Rabalais et al., 1999). The research results in this manuscript could provide guidance for selection of tile drainage routines and related parameter sets for tile drainage simulation at both field and watershed scales, to help solve systematic water quality issues in the old tile drainage systems. For example, well calibrated routines and related parameter sets in this study have been used for modeling of the impacts of bioenergy crop scenarios on streamflow, tile flow, sediment and nitrate losses in the LVR watershed from 1990 to 2008 (Guo *et al.*, 2018), to help determine optimal bioenergy scenarios with high biomass yields, and water quality benefits in the LVR watershed and even the Mississippi River system and Gulf of Mexico. Additionally, the calibrated tile drainage parameters also have provided guidance to modeling study on the impacts of conservation practices on crop yields and nutrient reductions at USDA-ARS edge-of-field sites and other agricultural fields using the Nutrient Tracking Tool (NTT), a web-based frontend of the Agricultural Policy/Environment eXtender (APEX) in northwestern Ohio (Guo et al., 2017, unpublished), to achieve the nutrient reduction goal in Lake Erie. Moreover, research results also provided guidance to SWAT model setup and calibration for the Maumee Basin, in a project on a multi-modeling approach to help policymakers determine potential solutions to elevated phosphorus loads, and consequently harmful algal blooms (HABs) in Lake Erie (Communications with Five SWAT groups, Sep 2017). (please see related discussion in Introduction, page 3, line 9-29).

References:
Guo, T., Raj, C., Chaubey, I., Gitau, M., Arnold, J. G., Srinivasan, R., Kiniry, J. R. & Engel, B. A. (2018). Evaluation of bioenergy crop growth and the impacts of bioenergy crops on streamflow, tile drain flow and nutrient losses in an extensively tile-drained watershed using SWAT. Science of the Total Environment, 613–614 (2018) 724–735. https://doi.org/10.1016/j.scitotenv.2017.09.148

Communications with Five SWAT groups, HABRI Stakeholder Advisory Group Meeting. September 29, 2017, Heidelberg University, Tiffin, OH, USA.

Guo, T., Confesor, R., Saleh, A., & King, K. (2017). Evaluation and application of Nutrient Tracking Tool in Northwestern Ohio (manuscript).

Jaynes, D., James, D., 2007. The extent of farm drainage in the United States. Annual Meeting of the Soil and Water Conservation Society, Tampa, Florida.

Kalita, P.K., Cooke, R.A.C., Anderson, S.M., Hirschi, M.C., Mitchell, J.K., 2007. Subsurface drainage and water quality: the Illinois experience. Trans. ASABE 50 (5), 1651–1656.

Rabalais, N., Turner, R., Justic, D., Dortch, Q., Wiseman, W., 1999. Characterization of hypoxia. Topic 1 report for the integrated assessment of hypoxia in the northern Gulf of Mexico. NOAA Coastal Ocean Program Decision Analysis Series Report No. 15.

**Comments from Referee # 1:** Some of the parameter values are also hard to understand, for instance, the range of snow fall and snow melt parameters seems too large (-5 to 5 °C). From physical process point of view, it is hard to explain why these parameters are so different in

such a small and mildly sloped watershed? To mention another example, why fitting values of SURLAG differ between sites (how scaling in hydrology may guide explaining this?). Similar can be said for other parameters like curve number, sediment and nitrogen related parameters. Therefore, the currently presented parameter sets adds very little to the available knowledge. A critical discussion on the fitted parameter values, at least explaining physical process related reasons and issues of spatial scales, is recommended.

*Response***:** Yes, we agree that the range of snow fall and melt parameters were large, and calibrated parameters (eg., SURLAG) should not have large differences between sites. Thus, we recalibrated/revalidated the model at all sites. We completed multi-objective calibration of SWAT with the old and new routines using the R language. We excluded snow, groundwater and soil water process parameters and soil and site properties related parameters in calibration to avoid overcalibration (Table 2), and focused on high influence parameters determined from the previous calibration/validation using SWAT-CUP and the previous studies in the region (Boles et al., 2015; Moriasi et al., 2012; Moriasi et al., 2013), in order to obtain realistic parameter sets to represent physical processes. Given that surface runoff rarely occurred in the LVR, we also changed parameter range for SURLAG from "0.5-2.0" to "0.0-2.0", to decrease the portion of the surface runoff release to the main channel. Generally, we obtained acceptable calibrated results with similar parameter sets in reasonable ranges at different sites (Please see Table 3).

Land use, soil, weather, patterns of tile drainage systems, and management practices are different at different stations, and thus it is reasonable to have different calibrated parameter sets for tile drainage simulation. However, parameter sets are realistic and similar to each other at different stations (Please see Table 3).

We would like to thank referee # 1 for the suggestions about a critical discussion on the fitted parameters. We have incorporated more in-depth discussion about how the calibrated parameter sets for different routines represent the physical processes of tile drainage for each indictor (please see section 3 Results and discussion). For example, for the old tile drainage routine in Rev.528 parameters, we discussed "The calibrated TDRAIN values were 26 and 25 hs for sites B and E, representing that it would take 26 and 25 hs to drain soils from saturation to field capacity at sites B and E, respectively (Table 4). The calibrated drain lag time (GDRAIN) values were 25 and 26 hs for sites B and E, representing that there were 25 and 26 hrs lag time between water enters the tiles from soil and water enters the main channel from the tiles at sites B and E, respectively, which was used to smooth the tile flow hydrograph (Table 4). However, using a draindown time (TDRAIN) to determine tile flow rate was too simplified, since TDRIAN was a static value for tiles no matter whether there was a large storm or not." (please see page 12, lines 1-7). The calibrated parameters about the new routine in Rev.615 and Rev.645 and the

[revised manuscript text omitted]

**Comments from Referee # 1**:Another major problem is difficulty in following the structure of the paper. Presentation of calibration and validation results for each site demonstrates lot of repetition. This obstruct clarity and the readers could soon start feeling bored as same information comes again without any new insights and deeper discussion. One way of rectifying this issue could be by fully restructuring the paper. For example, results can be separately presented for each indicator (crop yields, flows, sediment, and nitrate) rather than per site. This can also facilitate physical explanation and scale issues when results of all sites for one indicator are combined together. For instance, when it comes to peak flow or runoff simulations, one can see where it was simulated well, at R5 or B or E etc, and then what could be the governing factors (geography, tile drainage density, variation in hydraulic conductivity, effect of CN etc).

*Response*: We thank referee #1 for the constructive suggestions about improving the structure of the manuscript. We have reorganized the results and discussion and present results for each indicator, to avoid repetition and improve flow of the manuscript (please see section 3 Results and Discussion). We also related parameter sets with physical processes of tile drainage, and compared the performance of the old and new routines in simulating the same indicator at different sites (please see section 3 Results and discussion).

We also discussed the reasons for the performance of the old and new routines in simulating results and related it to soil and weather characteristics, tile patterns and cropping systems of sites, and physical bases of routines and parameters. For example, we discussed "The old routine in Rev.528 had different performance at sites B and E, which was mainly caused by different soil and weather characteristics, tile pattern and cropping systems of the two sites (Table 1), and the physical process of simulating tile flow

in the old routine. For instance, site B had clay silt loam soil, random tile pattern and reduced-tillage practices, while site E had silt loam soil, constant tile spacing and no-tillage practices (Table 1). The old routine in Rev.528 has the potential to overestimate tile flow peaks, since simulated tile flow by the old routine was controlled by a simple drawdown time parameter (TDRAIN), and tiles were allowed to carry an unlimited maximum of water no matter how intense the rainfall. The calibrated TDRAIN values were 26 and 25 hs for sites B and E, representing that it would take 26 and 25 hs to drain soils from saturation to field capacity at sites B and E, respectively (Table 4). The calibrated drain lag time (GDRAIN) values were 25 and 26 hs for sites B and E, representing that there were 25 and 26 hrs lag time between water enters the tiles from soil and water enters the main channel from the tiles at sites B and E, respectively, which was used to smooth the tile flow hydrograph (Table 4). However, using a draindown time (TDRAIN) to determine tile flow rate was too simplified, since TDRIAN was a static value for tiles no matter whether there was a large storm or not. Thus, the old routine overestimated tile flow peaks for site E (Figs. 3c and 3d), which was consistent with tile flow simulation using the old routine in the Matson Ditch watershed in Indiana (Boles et al., 2015). Moreover, the old routine was used to simulate tile flow on days when the simulated height of the water table exceeded the height of the tile drain (Neitsch et al., 2011). Tile drainage systems can cause water table recession in tile-drained soil. Water table was lower when respiratory activity was highest in summer (Muhr et al., 2011), which may be lower than the depth of subsurface tiles during long dry summer periods. Water table depth calculation based on change in the soil water for the whole soil profile tended to overestimate the distance between water table and the soil surface when long-term simulations were performed, most commonly in cases where days without rainfall dominated (Moriasi et al., 2013). Thus, Rev.528 simulated tile flow was zero during long dry summer periods." (please see page 11, line 28 to page 12, line 15).

References:
Boles, C. M., Frankenberger, J. R., and Moriasi, D. N.: Tile Drainage Simulation in SWAT2012: Parameterization and Evaluation in an Indiana Watershed, Trans. ASABE, 58, 1201-1213, doi: 10.13031/trans.58.10589, 2015.
Moriasi, D. N., Gowda, P. H., Arnold, J. G., Mulla, D. J., Ale, S., Steiner, J. L., and Tomer, M. D.: Evaluation of the Hooghoudt and Kirkham tile drain equations in the Soil and Water Assessment Tool to simulate tile flow and nitrate-nitrogen, J. Environ. Qual., 42, 1699-1710, doi:10.2134/jeq2013.01.0018, 2013.
Muhr, J., Höhle, J., Otieno, D. O., and Borken, W.: Manipulative lowering of the water table during summer does not affect CO2 emissions and uptake in a fen in Germany, Ecol. Appl., 21, 391-401, doi:10.1890/09-1251.1, 2011.
Neitsch, S. L., Williams, J., Arnold, J., and Kiniry, J.: Soil andWater Assessment Tool Theoretical Documentation Version 2009, Grassland, Soil and Water Research Laboratory, Agricultural Research Service and Blackland Research Center, Texas Agricultural Experiment Station, College Station, Texas, 2011.

**Comments from Referee # 1:** Although the study mentions previous research on testing the new tile drainage routine, the results of this study are not compared with the previous findings. A detailed comparison with the previous studies would help to understand and position this work much better. While doing so, the authors should at least include topics related to parametrization, characteristics of the studied watersheds, performance evaluation results.

*Response*: Yes, we agree that comparison between the calibrated parameters, characteristics of study areas and model performance and the previous studies is needed. The previous study on testing the new tile drainage routine evaluated performance of the routine in simulating streamflow on a tile drained watershed, without observed tile flow data at field scales. We have compared our simulated results and performance evaluation results with the previous studies on simulation of crop yields, tile flow and nitrate concentration in tile flow using DRAINMOD and Root Zone Water Quality Model (RZWQM) at field scales in the LVR. We also compared values of the calibrated parameters with the previous studies on tile drainage simulation using SWAT in other watersheds in the Midwest.

[revised manuscript text omitted]

**Comments from Referee # 1**:Additionally, some very useful comments are made by S. Mylevaganam. In general, I see them valid and constructive (though critical) and could be helpful for improving the manuscript.

*Response*: We have considered the comments from Dr. Mylevaganam and improved the manuscript. Please see our responses to Dr. Mylevaganam's comments below:

**Interactive comments # 1**:

4) From the reader's point of view, the introduction of the manuscript needs to rewritten. In the current version of the manuscript, the introduction is built with many equations. From the reader's point of view, a section with all these equations need to be introduced after the introduction. This will help the authors to have an introduction to highlight the need of the research.

*Response*: We have moved the equations to section 2.1 Tile drainage routines in SWAT. The introduction has been reorganized to focus on the importance of the research.

**Interactive comments # 1**:

5) From the reader's point of view, some of the paragraphs in the introduction are not coherent.

*Response*: We have removed information not coherent in the introduction. We have rewritten the introduction to better focus on the importance of the research.

**Interactive comments # 1**:

6) From the reader's point of view, the conclusions need to be re-written. Some of the words (e.g., site B, site E, and R5) in the current version of the paper need to be deleted. The actual locations of the sites need

to be mentioned in the conclusion.

*Response*: We have deleted site name and described characteristics of the sites in the conclusions.

**Interactive comments # 1:**

7) In the abstract, the authors claim that both the routines provided reasonable but unsatisfactory uncalibrated flow and nitrate loss results. The authors should clearly state the meaning of "reasonable but unsatisfactory". Moreover, the authors need to state the temporal scale of their statement.

*Response*: "Both routines provided reasonable but unsatisfactory uncalibrated flow and nitrate loss results." has been changed to "Both the old and new routines provided reasonable but unsatisfactory (NSE < 0.5) uncalibrated flow and nitrate loss results for a mildly-sloped watershed with low runoff." (please see page 1, line 20-21).

**Interactive comments # 1:**

8) In the abstract, the authors claim that the new routine provided acceptable simulated tile flow and nitrate in tile flow for both field sites with random pattern tile and constant tile spacing. However, in the current version of the paper, the reader is unable to find more detail about the random pattern. Moreover, it would be more meaningful if the authors relate these patterns to the adopted equations shown in equations (3-5).

*Response*: The selected sites incorporated both random pattern tile and constant tile spacing. However, random tile spacing is still represented as a constant tile spacing in the model currently. As we mentioned, "There is an opportunity to improve the representation of tile drainage systems in SWAT, especially for individual tiles" (Please see page 18, lines 28-29). We believe that better representation of size and spatial information of tile drainage systems can improve simulation of tile drainage.

**Interactive comments # 1:**

9) In the current version of the paper, it is understood that there exists a coefficient named "drainage coefficient" (DC in equation-5) in SWAT 2009 and SWAT 2012. The authors also state that a coefficient named "drainage coefficient "(DRAIN_CO) was included in the new tile drainage routine in SWAT2012. Does SWAT2012 in its tile drainage routine have two drainage coefficients?

*Response*: No, DC and DRAIN_CO are the same. We have improved the description to be consistent.

**Interactive comments # 1:**

10) The authors need to clearly state the difference between SWAT2012 Rev.615 and SWAT2012 Rev.645.

*Response*: Compared to SWAT2012 Rev.615, SWAT2012 Rev.645 incorporated a new retention parameter adjustment factor (R2ADJ) to modify the soil moisture retention parameter calculation method. R2ADJ was used to modify shape coefficients, and curve number was calculated from capacity to

saturation. This method is more reasonable than decreasing curve number directly (please see page 6, line 23 to page 7, line 15).

**Interactive comments # 1:**

5  11) As per the current version of the paper, a coefficient named drainage coefficient (DRAIN_CO) was included in the new tile drainage routine in SWAT2012 to "control "peak drain flow. However, in the current version of the paper, the old tile drainage routine in SWAT2009 (Rev.528) and the new tile drainage routine in SWAT2012 (Rev.615 and Rev.645) were used to simulate monthly tile flow, nitrate in tile flow, surface runoff, and sediment and nitrate in surface runoff at field sites, and monthly flow,
10  sediment and nitrate in flow at a river station. Therefore, it is unclear about the motivation of this research work. Moreover, it would be meaningful if the authors show the equation that uses DRAIN_CO.
*Response*: The old routine in Rev.528 has the potential to overestimate tile flow peaks, since simulated tile flow by the old routine was controlled by a simple drawdown time parameter (TDRIAN), and tiles were allowed to carry an unlimited maximum of water. Thus, Rev.528 has the potential to overestimate
15  tile flow peaks and nitrate in tile flow at site E. On the contrary, simulated tile flow peaks and nitrate in tile flow peaks from the new routine in Rev.615 and Rev.645 captured the observed values fairly well. The motivation of this research is to compare the performance of different tile drainage routines in simulating water quantity and quality at field and watershed scales, and determine the most suitable model for further simulation in the extensively tile-drained watershed. The drainage coefficient represents the
20  maximum amount of water that can be drained from tiles. The tile flow is set equal to the drainage coefficient once tile flow is greater than drainage coefficient (please see equations on page 5, lines 9-16). We also described "(Boles et al., 2015). DRAIN_CO, the amount of water drains in 24 hs, was set as 20 mm day-1, describing the size of the main collector drain pipes and the outlet (Sui and Frankenberger, 2008)." (please see page 8, lines 19-21).

**Interactive comments # 1:**
12) The Fig 1 needs to be checked by a GIS professional. From the reader's point of view, the Fig 1 is meaningless. Moreover, there is an asterisk within the IL boundary. This asterisk should be related to the main figure. The abbreviation "Co." is not understood. The caption of the figure needs to be self-
30  illustrative. The county borders also need to be checked. Do they intersect orthogonally?
*Response*: We have improved Fig.1 to better present study area information. The asterisk within the IL boundary has been removed. The abbreviation "Co." has been changed to "County". The caption of Fig.1 has been changed to "Schematic of the LVR watershed with location of monitored subsurface, surface and river stations". We have created a new Fig. 1 using county borders downloaded as an ArcGIS
35  shapefile.

**Interactive comments # 1:**

13) In Fig 1, is the river station R5 shared by both the counties (i.e., Vermillion and Champaign counties)?

*Response*: Yes, the river station R5 is on the county line and shared by both counties.

14) The authors need to state few lines about the methodology used to get the drainage areas of subsurface stations and surface runoff stations.

*Response*: The drainage areas of subsurface and surface stations were determined from the hand drawn tile layout with locations of tile lines and monitoring stations at the study sites in the LVR. Please see Fig. 2 in (Singh et al., 2001)'s study. Sites B and F in Fig. 2 below represent sites B and E in current study, respectively. We have added this description in section 2.2 (please see page 6, lines 5-7).

Figure. 2. Tile layout at various study sites in the LVR watershed.

Rev.528 simulated tile flow was zero during long dry summer periods." (please see page 11, line 28 to page 12, line 15). We also discussed simulated results and parameters for the new routine in the following paragraph.

**iv)** The area covered by surface and sub-surface station is as low as in range of 0.05 km2. What is HRU size corresponding to drainage area for B and E? This information will reveal how well drainage is simulated in the considered drainage area.
*Response*: Yes, we have the same concern. HRU size in SWAT is 14.18 and 0.72 km$^2$, respectively. HRU size in SWAT is larger than the size of the station. As we mentioned in the Limitation section, there is an opportunity to improve the representation of tile drainage systems in SWAT, especially for individual tiles. We believe that better representation of size and spatial information of tile drainage systems can improve simulation of tile drainage.

**v)** Leave-few-year out approach may be more suitable for calibration and validation.
*Response*: We have recalibrated models for monthly water quantity and quality results at all fields using leave-few-year out approach, which was suitable for our study. We used monthly results during the first 8-10 years (varies at different sites) for calibration and during the last two years for validation. For annual crop yields, we still used 7 years of data for calibration and 6 years of data for validation, since leave-few-year out approach would make sample points two few (two annual data) for validation (please see page 9, lines 4-11).

**vi)** Introduction can be reconstructed. In current form science question are repeated at two places on page 2 line 5 and page 4 line 32.
*Response*: We thank referee # 2 for the detailed suggestions to the structure and description of this manuscript. We are very grateful. The Science question on page 2 line 5 has been removed. The introduction has been reorganized to improve the flow of the manuscript and to better focus on the importance of the research.

**vii)** (line 20) Explanation is required on how uncalibrated routines give 'reasonable but unsatisfactory' performance.
*Response*: "Both routines provided reasonable but unsatisfactory uncalibrated flow and nitrate loss results." has been changed to "Both the old and new routines provided reasonable but unsatisfactory (NSE < 0.5) uncalibrated flow and nitrate loss results for a mildly-sloped watershed with low runoff." (please see page 1, lines 21-22).

**viii)** Page 5 line 25 citation is improper
*Response*: Citation on page 5 line 30 has been corrected to Mitchell et al. (2003) and Kalita et al. (2006).

**ix**) Page 9 line 27 variables of equation are not properly defined.

*Response*: We have changed "Where Obs and Sim represent observed and simulated data, respectively." to "Where *Obs* and *Sim* represent the ith observed and simulated monthly data, respectively. n is the total number of months. $\overline{Obs}$ and $\overline{Sim}$ represent the average values of the observed and simulated monthly data, respectively." (please see page 10, lines 3-4).

**x)** Repetition: Page 14 line 13-14, Two sentences can be merge in 1. Page 14-19 looks like repetition of sentences.

*Response*: The sentence on page 14 line 13-14 has been condensed to "Performance of the modelled monthly surface runoff at sites Bs and Es during the calibration and validation was satisfactory from Rev.645, and was unsatisfactory from Rev.615. " (please see page 13, lines 29-30). The sentences from page 14 to 19 have been reorganized to present results for each indicator, to avoid repetition and improve flow of the manuscript.

**xi)** Page 18 line 31, 'both routines' which two? Is not clear.

*Response*: Both routines represented the old tile drainage routine in SWAT2009 (Rev.528) and the new tile drainage routine in SWAT2012 (Rev.615 and Rev.645), which was mentioned on page 18, lines 31-32. We have changed 'both routines' to 'both the old and new routines'.

The list of changes made in the manuscript:

1. We developed a multi-objective calibration tool for SWAT model in the R rather than SWAT-CUP for recalibration.

2. We recalibrated the models at all sites based on the improved parameters selected for calibration, and an improved calibration/validation periods.

3. We moved the equations about tile drainage routines from the introductions to the materials and methods.

3. We rewrote the introduction to better focus on the importance of the research.

4. We reorganized the results and discussion to discuss the simulated results based on each indicator rather than each site.

5. We incorporated in-depth discussion about the calibrated parameters and tile drainage routines related to physical processes.

6. We compared the calibrated parameters and model performance with the previous studies.

These changes are shown by track changes in the marked-up manuscript version.

[revised manuscript text omitted]
 | 0.01~4 | - | 1 | 1.0 | - | 1.00 | 1.00 | 1.41 | 1.40 | 1.21 | 1.20 | - | 1.1 |
|  | ~~25000~50000330003700028000280003600029000290004100038000~~ |
|  | ~~0.8~0.8-0.240.68-0.790.32-0.620.620.030.52-0.170.36-0.260.07~~ |
| ESCO | 0.80~0.99 | 0.88 | 0.91 | 0.91 | 0.90 | 0.88 | 0.91 | 0.91 | 0.88 | 0.90 | 0.90 | 0.91 | 0.90 |
|  | ~~-5~5-1.792.77-4.47-1.991.343.35-4.964.374.533.970.58-4.25~~ |
|  | ~~-5~5-2.282.59-3.783.390.86-1.52-1.44.80.111.570.992.08~~ |
|  | ~~10~40162922272120121632193725~~ |
|  | ~~0~0.30.050.090.280.040.110.030.080.200.210.280.720.56~~ |
|  | ~~0.2~0.20.05-0.190.180.04-0.090.030.03-0.160.190.150.06-0.03~~ |
| ADJ_PKR | 0.5~2 | - | - | - | - | - | - | - | 1.2 | 1.1 | - | 1.2 | 1.2 |
| USLE_C {19} | -0.25~0.25 | - | - | - | - | - | - | - | 0.12 | 0.15 | - | 0.15 | 0.15 |
| USLE_C {56} | -0.25~0.25 | - | - | - | - | - | - | - | 0.05 | 0.06 | - | 0.07 | 0.07 |
|  | ~~0~0.0200.020.020~~ |
|  | ~~0.1~0.10.080.03~~ |

| Parameter | Range | | | | | | | | | | | | |
|---|---|---|---|---|---|---|---|---|---|---|---|---|---|
| USLE_K(1) | 0.1~0.1 | - | - | - | - | - | - | - | 0.07 | - | 0.1 | 0.1 | -0.06 |
| SPEXP | 1~2 | - | - | - | - | - | - | - | - | - | - | 1.50 | 1.945 |
| CH_COV1 | 0~1 | - | - | - | - | - | - | - | - | - | - | 0.38 | 0.31 |
| CMN | 0.0003~0.03 | 0.02 | 0.02 | - | 0.000302 | 0.02 | - | - | 0.031 | - | 0.032 | 0.0003 | 0.03 |
| RCN (mg N L$^{-1}$) | 0~15 | 911 | 915 | - | 110 | 11 | - | - | 96 | - | 105 | 911 | 90.1 |
| NPERCO | 0~1 | 0.1584 | 0.1501 | - | 0.1253 | 0.4812 | - | - | 0.9912 | - | 10.13 | 0.9915 | 0.9915 |
| SDNCO | 0~1.5 | 1.125 | 1.126 | - | 1.102 | 1.139 | - | - | 1.300.9 | - | 0.93 | 1.0 | 1.046 |
| CDN | 0~1 | 0.061 | 0.062 | - | 0.3305 | 0.2805 | - | - | 10.06 | - | 10.05 | 0.06 | 0.06 |

Negative value for CN II CN2, and value for SOL_K(1), SOL_AWC(1) (mm H$_2$0 mm$^{-1}$ soil), USLE_K(1) (0.013 (metric ton m$^2$ h)/(m$^3$ metric ton cm)), USLE_C{19}, and USLE_C{56} is relative change to default value. (1) indicates the first soil layer. {19} and {56} represent corn and soybean, respectively.

**Table 4** Performance evaluation of the calibrated and validated results at sites B, E, Bs, Es and R5

| | Annual Crop yield (t ha⁻¹) | | Monthly Tile flow (mm) | | | | | | Monthly NO3-N in tile flow (kg ha⁻¹) | | | |
|---|---|---|---|---|---|---|---|---|---|---|---|---|
| | Cali | Vali | Cali | | | Vali | | | Cali | | Vali | |
| Revision | 615 | 615 | 528 | 615 | 645 | 528 | 615 | 645 | 528 | 615 | 528 | 615 |
| | | | | | Site B | | | | | | | |
| P_BIAS (%) | 13 | 2 | 3 |  | 28 | 1 |  | -18 | 9 | 24 | 31 | 34 |
| R² | 0.99 | 0.92 | 0.73 | 0.49 | 0.726 4 | 0.80 | 0.69 | 0.64 | 0.65 | 0.377 0 8 | 0.677 6 | 0.787 6 |
| NSE | 0.91 | 0.91 | 0.71 | 0.54 | 0.666 1 | 0.80 | 0.68 | 0.58 | 0.66 | 0.226 8 | 0.637 5 | 0.776 8 |
| MSE | 0.77 | 0.76 | 0.60 | 0.54 | 0.534 9 | 0.67 | 0.53 | 0.48 | 0.59 | 0.436 0 | 0.556 1 | 0.645 0 |
| KGE | 0.75 | 0.89 | 0.85 | 0.70 | 0.756 3 | 0.86 | 0.78 | 0.73 | 0.68 | 0.487 4 | 0.716 9 | 0.786 8 |
| | | | | | Site E | | | | | | | |
| P_BIAS (%) | -2 | 5 | 24 | -3 | 17 | 49 | 12 | -2 | 4 | 28 | 85 | 20 |
| R² | 0.95 | 0.92 | 0.68 | 0.50 | 0.586 | 0.75 | 0.55 | 0.56 | 0.72 | 0.60 | 0.385 0 | 0.56 |
| NSE | 0.95 | 0.88 | -0.77 | 0.50 | 0.535 4 | -0.2 | 0.48 | 0.53 | -0.09 | 0.51 | 0.21 | 0.48 |
| MSE | 0.80 | 0.71 | -0.20 | 0.28 | 0.272 8 | 0.04 | 0.32 | 0.39 | 0.08 | 0.32 | 0.18 | 0.313 4 |
| KGE | 0.95 | 0.91 | -0.05 | 0.61 | 0.717 1 | 0.15 | 0.65 | 0.76 | 0.24 | 0.686 6 | 0.45 | 0.576 5 |

| | Monthly Surface runoff (mm) | | | | Monthly Sediment (t ha⁻¹) | | Monthly Nitrate in runoff (kg ha⁻¹) | |
|---|---|---|---|---|---|---|---|---|
| | Cali | | Vali | | Cali | Vali | Cali | Vali |
| Revision | 615 | 645 | 615 | 645 | 645 | 645 | 645 | 645 |
| | | | | Site Bs | | | | |
| P_BIAS (%) | 108 | 13 | 143 | 12 | 20 | -77 | 99 | 77 |
| R² | 0.23 | 0.88 | 0.76 | 0.80 | 0.96 | 0.13 | 0.14 | 0.01 |
| NSE | -4.70 | 0.81 | -5.95 | 0.64 | 0.95 | 0.11 | 0.06 | -0.32 |
| MSE | -2.36 | 0.48 | -1.70 | 0.41 | 0.74 | 0.48 | 0.46 | 0.54 |
| KGE | -5.33 | 0.58 | -4.22 | 0.18 | 0.86 | 0.55 | -0.34 | -0.11 |
| | | | | Site Es | | | | |
| P_BIAS (%) | 135 | 18 | 82 | 28 | -8 | 86 | 70 | 59 |
| R² | 0.71 | 0.65 | 0.55 | 0.48 | 0.79 | 0.11 | 0.33 | 0.005 |
| NSE | 0.50 | 0.50 | -0.85 | -0.28 | 0.46 | 0.10 | -0.13 | -0.07 |
| MSE | 0.28 | 0.42 | 0.01 | 0.71 | 0.07 | 0.50 | 0.28 | 0.35 |
| KGE | -0.44 | 0.64 | -0.67 | -0.15 | 0.78 | 0.22 | 0.78 | -0.54 |

| | Monthly Flow (cms) | | | | Monthly Sediment (t) | | | | Monthly Nitrate (kg) | | | |
|---|---|---|---|---|---|---|---|---|---|---|---|---|
| | Cali | | Vali | | Cali | | Vali | | Cali | | Vali | |
| Revision | 528 | 615 | 528 | 615 | 528 | 615 | 528 | 615 | 528 | 615 | 528 | 615 |
| | | | | | Site R5 | | | | | | | |
| P_BIAS (%) | 16 | 26 | 1 | 37 | 44 | -55 | -19 | 3004 | 27 | 31 | 6 | 7 |
| R² | 0.85 | 0.84 | 0.91 | 0.63 | 0.43 | 0.40 | 0.96 | 0.95 | 0.59 | 0.63 | 0.70 | 0.68 |
| NSE | 0.77 | 0.73 | 0.89 | 0.48 | 0.40 | 0.45 | -1.05 | -9.67 | 0.43 | 0.58 | 0.57 | 0.65 |
| MSE | 0.50 | 0.43 | 0.67 | 0.26 | 0.49 | 0.46 | 0.07 | -1.77 | 0.41 | 0.50 | 0.51 | 0.57 |

| KGE | 0.6375 | 0.5067 | 0.8078 | 0.671 | 0.2800 1 | 0.56- 0.13 | - 0.6323.48 | - 4.6135.24 | 0.625 | 0.7153 | 0.6479 | 0.648 |

Cali and Vali represent calibration and validation, respectively.

**Fig. 1**

[Figure]

**Fig. 2**

[Figure]

**Fig. 3**

[Figure]

**Fig. 4**

[Figure]

**Fig. 5**

[Figure]

**Fig. 6**

[Figure]

**Fig. 7**

[Figure]